



# Potential modulation of Indian Ocean basin mode on the interdecadal variations of summer precipitation over the East Asian monsoon boundary zone

Jing Wang[1,*], Yanju Liu[2], Fei Cheng[3,*], Chengyu Song[4], Qiaoping Li[5], Yihui Ding[2], Xiangde Xu[6]

[1]Tianjin Key Laboratory for Oceanic Meteorology, and Tianjin Institute of Meteorological Science, Tianjin, China

[2]National Climate Center, China Meteorological Administration, Beijing, China

[3]Ningbo Meteorological Observatory, Ningbo, China

[4]Heilongjiang Climate Centre, Harbin, China

[5]CMA Earth System Modelling and Prediction Centre, Beijing, China

[6]State Key Laboratory of Severe Weather, Chinese Academy of Meteorological Sciences, Beijing, China

Correspondence: Yanju Liu (liuyanj@cma.gov.cn)
* Jing Wang and Fei Cheng contributed equally to this work.

**Abstract.** Based on long-term observational and reanalysis datasets from 1901 through 2014, this study investigates the characteristics and physical causes of the interdecadal variations in the summer precipitation over the East Asian monsoon boundary zone (EAMBZ). Observational evidence reveals that the EAMBZ precipitation featured prominent interdecadal fluctuations, e.g., with dry summers during the periods preceding 1927, 1968–1982, and 1998–2010, and wet summers during the periods of 1928–1938, 1946–1967, and 2011 onwards. Further analyses identify that the Indian Ocean basin mode (IOBM) is an important oceanic modulator responsible for the interdecadal variations of the EAMBZ precipitation. When the cold phase of the IOBM occurs, an anomalous cyclonic circulation is excited around the northeast corner of the tropical Indian Ocean, which further induces a "north-low–south-high" meridional seesaw pattern over the Northeast China–subtropical western Pacific (SWP) sector. Such seesaw pattern is conducive to the enhanced EAMBZ precipitation through linking favorable environments for the transportation of water vapor from the SWP and the convergence over EAMBZ at interdecadal timescales. For this reason, a physical-empirical model for the EAMBZ precipitation is developed in terms of the IOBM cooling, which can well capture its interdecadal fluctuations and reflect their steady relationship. The key physical pathway connecting the IOBM cooling with the interdecadal variations of the summer EAMBZ precipitation is supported by the numerical results based on the large ensemble experiment and the Indian Ocean pacemaker experiment. Our findings may provide new insights into the understanding of the causes of the interdecadal variations in the summer EAMBZ precipitation, which may favor the long-term policy decision making for the local hydrometeorological planning.

## 1 Introduction

The monsoonal airflows and mid-latitude westerlies are crucial components of the Asian climate system (Li and Zeng, 2002; Ding and Chan, 2005; Wang et al., 2008; Wu et al., 2012; Huang et al., 2015; Wang et al., 2017; Chen et al., 2018; J. Huang et al., 2019). These two subsystems can synergistically induce regional precipitation fluctuations over subtropical and mid-latitude Asia during the Northern Hemisphere (NH) late spring (May) and summer (June–August; JJA) (Qian et al., 2009; Chen et al., 2021; Song et al., 2022; J. Wang et al., 2022). For example, Song et al. (2022) found that May precipitation over the southeastern extension of the Tibetan Plateau (TP) features notable year-to-year variations, which are physically linked to a unique interplay between the upstream mid-latitude westerlies and the Bay of Bengal summer monsoon.






During the early stage of the northern summer, however, the mid-latitude NH westerlies shift poleward to the north of the
TP abruptly (Yeh et al., 1959; Schiemann et al., 2009). In this context, westerlies of mid-latitude synoptic disturbance and
southerlies of East Asian summer monsoon (EASM) collide with each other frequently over the East Asian monsoon
boundary zone (EAMBZ) (Qian et al., 2009; Wang et al., 2017; Chen et al., 2018; J. Huang et al., 2019; Zeng and Zhang,
2019; Chen et al., 2021; Q. Wang et al., 2021, 2022, 2023). EAMBZ is a transitional climate zone between the EASM-
controlled moist region and the westerly-dominated arid region over central Asia (Chen et al., 2010; Chen et al., 2018,
2021), stretching from the eastern flank of the TP to Mongolia and Northeast China [see Fig. 1 in Chen et al. (2021); also
see the red box in **Fig. 1**]. Notably, EAMBZ is a distinguished region with agrarian economy and animal husbandry, which
is largely susceptible to water resource variations (Ou and Qian, 2006; Lu and Jia, 2013). Nevertheless, many studies
reported that in the past century, the semi-arid EAMBZ underwent the most profound warming over East Asia, suffering
from serious aridification and a high risk of desertification (J. Huang et al., 2017, 2019, 2020). In this regard, EAMBZ is
deemed one of the "hotspots" highly sensitive to precipitation fluctuations (Qian et al., 2009; Lu and Jia, 2013; J. Huang
et al., 2019). Given that the EAMBZ is of an ecologically fragile environment with water shortage, a deep understanding
of the reasons for historical changes in summer EAMBZ precipitation could be a prerequisite for in situ ecological
improvement and socioeconomic development.

Existing studies have well documented physical mechanisms responsible for the interannual variability of summer
EAMBZ precipitation, highlighting the external moisture supply pathways, the modulators for the wet-dry condition
variations [e.g., the mid-latitude westerlies within the Asian westerly jet (AWJ), the western North Pacific (WNP)
subtropical high (WNPSH), and the EASM], and the remote modulation roles of large-scale teleconnected modes [e.g.,
Silk Road pattern (SRP)/circumglobal teleconnection (CGT) propagating along the AWJ and the Eurasian teleconnection
(EU)] and sea surface temperature (SST) anomaly patterns (Huang et al., 2015; Wang et al., 2017; Chen et al., 2018, 2021;
Zhao et al., 2019a, 2019b, 2020; Q. Wang et al., 2021, 2022, 2023). For instance, Q. Wang et al. (2022) suggested that
the positive EU phase is connected with a low pressure anomaly in the lower troposphere in EAMBZ and the Mongolia
region, thus favoring enhanced summertime precipitation over EAMBZ; and meanwhile, the CGT is positively coupled
with the EAMBZ precipitation, with ascending motion anomalies over EAMBZ during its positive phase. Chen et al.
(2021) established that the circulations (i.e., the mid-latitude westerlies and EASM) and the forcing of SST anomalies
(SSTAs) can collectively regulate the summer EAMBZ precipitation variability. The variability of westerlies is largely
modulated by the SRP and the meridional displacement of AWJ; while the EASM variability is mainly modulated by El
Niño-Southern Oscillation (ENSO). The synchronized effects of EASM and westerlies largely contribute to the rainfall
variability in EAMBZ. Zhao et al. (2019a) found that the tropical northern Atlantic SSTAs have significant impacts on
the August rainfall over the monsoon transitional zone in China through inducing a wavetrain over Eurasia and an
anomalous WNPSH.

Compared with the extensively explored interannual variability of the JJA EAMBZ precipitation, less efforts have been
devoted to its interdecadal variability. To understand and predict the summer EAMBZ precipitation, exploring its
interdecadal variations and the underlying physical causes are also critical, which are the main focus of the present study.
Previous studies suggested that the warm-season precipitation over many Asian areas features interdecadal fluctuations,
such as the southeastern TP in late spring (J. Wang et al., 2022) and East/Northeast Asia in summer (Si and Ding, 2016;
Zhang et al., 2018; Sun et al., 2019a; Piao et al., 2021). The oceanic interdecadal signals for these interdecadal changes
are also extensively investigated, highlighting the crucial modulation roles of basin-scale SST modes of Atlantic
multidecadal oscillation (AMO), Pacific decadal oscillation (PDO)/interdecadal Pacific oscillation (IPO), and Indian



Ocean basin mode (IOBM) (Si et al., 2021). Among these forcings, it is essential to emphasize the IOBM, a dominant
mode of SST variability in the tropical Indian Ocean (TIO) sector, which usually follows up a wintertime ENSO event
and persists into the summer through the capacitor effect (Klein et al., 1999; Yang et al., 2007; Xie et al., 2009). The
EASM is simultaneously correlated with the IOBM in boreal summer, which can be considered as a salient modulator for
the summer EAMBZ precipitation variability on interannual timescales (Chen et al., 2021). It is worth noting that the
IOBM also features a basin-scale warming/cooling at interdecadal timescales (Han et al., 2014), exerting active impacts
on the mid-latitude Asian climate (e.g., Wu et al., 2016; Li and Ma, 2018; Zhang et al., 2018; S. Wang et al., 2022). As
for the interdecadal variations of the summer EAMBZ precipitation, we hope to answer the following two questions: 1)
Did the JJA EAMBZ precipitation feature interdecadal variations? If so, 2) is there any intimate connection between
IOBM and the EAMBZ precipitation at interdecadal timescales? As such, this study shall extend previous studies by
exploring what extent and how the JJA IOBM modulate the concurrent EAMBZ precipitation variability at interdecadal
timescales, with the aim of providing a novel understanding for the rainfall variability over the mid-latitude semi-arid
zone in Asia. Note that we employ datasets with a centennial scale in this study [e.g., the precipitation data produced by
the Climatic Research Unit (CRU) and the atmospheric circulation data from the Twentieth Century Reanalysis (20CR)
datasets]. In comparison with the short-term datasets since the latter half of the 20th century, these long-term datasets can
separate the interdecadal variability of EAMBZ precipitation from the externally forced global climate change caused by
anthropogenic (e.g., greenhouse gases) and natural forcings (e.g., volcanic eruptions) more effectively (Wu et al., 2016),
which were widely used to investigate the physical causes of how internal fluctuations of the climate system modulate
the interdecadal variations of precipitation over Asia (e.g., Wu et al., 2016; Zhang et al., 2018; Sun et al., 2019a; Jiang et
al., 2021; J. Wang et al., 2022).

The remainder of this paper is arranged as follows. Section 2 describes the datasets and methods used in this study. Section
3 elucidates the characteristics of the interdecadal variations of summertime EAMBZ precipitation and the associated
background circulations, illustrates the mechanisms of how IOBM modulates the EAMBZ precipitation, and establishes
a linear regression model using the IOBM to predict the interdecadal precipitation anomalies over EAMBZ. Further
discussion and a summary of the major findings are provided in Section 4 and Section 5, respectively.

## 2 Datasets and methods


### 2.1 Observational Data


Several monthly mean observational datasets are utilized in the present study, including (1) the global land high-resolution
gridded CRU time series (TS) precipitation dataset version 3.26 (CRU TS3.26; spatial resolution: 0.5 °×0.5 °; Harris et al.,
2014) for 1901–2017, (2) the Extended Reconstructed SST version 5 (ERSSTv5; spatial resolution: 2 °×2 °; B. Huang et
al., 2017) for 1854–present derived from the National Oceanic and Atmospheric Administration (NOAA), and (3)
atmospheric variables derived from NOAA–Cooperative Institute for Research in Environmental Sciences (CIRES) 20CR
version 2c (20CRv2c; spatial resolution: 2 °×2 °; Compo et al., 2011), except for the precipitation data, with 192 points in
longitude and 94 points in latitude, for 1851–2014. Note that all observational datasets cover the common time period of
1901–2014, which is the focused period in the present research.

### 2.2 Rossby wave source


Following Sardeshmukh and Hoskins (1988), the Rossby wave source (RWS) is calculated as:



$$RWS = -\nabla \cdot \left[ V_{\chi} (\zeta + f) \right], \qquad (1)$$
where $V_{\chi}$ is the divergent wind, $\zeta$ is the relative vorticity, and $f$ is the planetary vorticity.

### 2.3 Moisture flux and associated divergence


The vertically integrated horizontal water vapor transport ($\langle \text{WVT} \rangle$) and associated divergence ($\langle \text{WVT\_div} \rangle$) are
calculated using the following equations (Sun et al., 2019b; J. Wang et al., 2022):
$$\langle \text{WVT} \rangle = -\frac{1}{g} \int_{P_s}^{300} q\vec{V} dp, \qquad (2)$$
$$\langle \text{WVT\_div} \rangle = -\frac{1}{g} \int_{P_s}^{300} \nabla_p \cdot (q\vec{V}) dp, \qquad (3)$$
where $\nabla_p \cdot ()$ denotes the horizontal divergence in the pressure coordinates; $g$ is the gravitational acceleration; $P_s$ is the
surface pressure; $q$ is the specific humidity; and $\vec{V} = (u, v)$ is the horizontal wind vector ($u$ and $v$ represent the zonal and
meridional winds, respectively).

### 2.4 Statistical methods


This study focuses on interdecadal fluctuations in variables. The data are 11-year low-pass filtered by adopting a Lanczos
filter (Duchon, 1979) to extract the corresponding interdecadal signal. Several statistical methods are used, including
empirical orthogonal function (EOF) analysis, composite analysis, correlation analysis, and linear regression analysis. A
two-tailed Student's $t$ test is used to evaluate the statistical significance. Considering the 11-year low-pass filtered method
can significantly reduce the degrees of freedom of the data, the following approximation is therefore deployed to calculate
the effective degrees of freedom ($N^{\text{eff}}$):
$$\frac{1}{N^{\text{eff}}} \approx \frac{1}{N} + \frac{2}{N} \sum_{j=1}^{N} \frac{N-j}{N} \rho_{XX}(j)\rho_{YY}(j), \qquad (4)$$
where $N$ is the sample size, and $\rho_{XX}(j)$ and $\rho_{YY}(j)$ are the autocorrelations of two sampled time series $X$ and $Y$,
respectively, at time lag $j$ (Li et al., 2013).

In this study, we focus on the boreal summer season (JJA). All variables in observations and model simulations are
linearly detrended before further calculations and analyses to exclude potential impacts of long-term trends.

### 2.5 Definitions


#### 2.5.1 The research domain of EAMBZ


From the long-term (1901–2014) perspective of the climatological mean state of converged <WVT> and pronounced
precipitation over the mid-latitude Asia, the EAMBZ (box in **Figs. 1a** and **1b**; 35 °–55 °N, 105 °–130 °E) is defined as the
collision and convergence zone between JJA dry westerly <WVT> and moist southwesterly <WVT> (**Fig. 1a**). As such,
there exist wetter conditions over the EASM-dominated part and drier conditions over the westerly-controlled part (**Fig.**
**1b**), suggesting the semi-arid transitional feature of EAMBZ (Xing and Wang, 2017). Our defined research domain of
EAMBZ largely matches the monsoon boundary zone defined by Chen et al. (2021), covering Inner Mongolia, Gansu,



Ningxia, Shaanxi, Shanxi, Hebei, Beijing, Tianjin, Shandong, Jilin, Liaoning, and Heilongjiang in China, as well as
eastern Mongolia and Korean peninsula. Note that the areal mean precipitation over EAMBZ in boreal summer is the
highest of the year accompanying the largest standard deviation (i.e., largest rainfall variability) (**Fig. S1**), which is the
focused season in the present study.

**2.5.2 Climate indices**

The IOBM index ($I_{IOBM}$) is defined as areal mean SSTAs over the TIO domain of 20 °S–20 °N, 40 °–100 °E (Xie et al.,
2009). The IPO index is calculated using a method identical to that defined in Henley et al. (2015), that is, the difference
between SSTAs averaged over the central equatorial Pacific (10 °S–10 °N, 170 °E–90 °W) and the average of SSTAs in the
northwest (25 °–45 °N, 140 °E–145 °W) and the southwest Pacific (50 °S–15 °S, 150 °E–160 °W). In observations,
considering the coupled nature of IOBM and IPO at interdecadal timescales in boreal summer [cf. Fig. 2a in Wu et al.
(2016)], we hence remove the potential influence of the contemporaneous IPO on precipitation via eliminating the signal
of IPO index from the data of climate variables based on the partial regression technique, which is widely used in previous
studies (e.g., Dou and Wu, 2018; J. Wang et al., 2022).

**2.6 Model simulations**

To validate our proposed mechanisms of how the TIO SSTAs (i.e., IOBM-associated SSTAs) remotely modulate the
summer EAMBZ precipitation on interdecadal timescales, following the method of Zhang et al. (2019) and Yang et al.
(2020), we adopt monthly mean outputs from two experiments of the Community Earth System Model version 1 (CESM1),
which is a fully coupled Earth system model incorporating components of atmosphere, ocean, land, and sea ice (Hurrell
et al., 2013).

The first experiment is the CESM1 Large Ensemble Numerical Simulation (referred to as CESM1_LENS; Kay et al.,
2015). Among total 40 ensemble members in CESM1_LENS (Yang et al., 2020), we use the first 35 individual members
according to many previous studies (e.g., Touma et al., 2021; J. Wang et al., 2023), which were completed at the climate
modeling center of National Center for Atmospheric Research (NCAR). Note that all ensemble members in
CESM1_LENS were imposed with the same radiative forcing scenario (Taylor et al., 2012), with historical forcing for
1920–2005 and high-emission forcing scenario [i.e., Representative Concentration Pathway (RCP) 8.5] for 2006–2080
(Moss et al., 2010; Touma et al., 2021). The ensemble members were further generated with slightly differentiated
perturbations of atmospheric states (Kay et al., 2015; Touma et al., 2021). The second experiment is the CESM1 Indian
Ocean Pacemaker Ensemble Simulation (referred to as CESM1_IOPES), with 10 ensemble members (Zhang et al., 2019;
Yang et al., 2020). We adopt CESM1_IOPES to highlight the impact of SSTAs over the broader TIO domain (15 °S–15 °N,
African coast to 174 °E). For the convenience of subsequent calculations and analyses, the African coast is designated as
40 °E in this study, and a small change in the longitudes regarding the African coast may not affect the main results.

In light of Yang et al. (2020), by subtracting the CESM1_LENS ensemble mean from the CESM1_IOPES ensemble mean,
we can obtain responses of the climate system to the internal variability stemming from the time-varying TIO SSTAs,
distinguishing the impact of external radiative force changes from the intrinsic variability driven by TIO SSTAs. More
details about CESM1_LENS and CESM1_IOPES can be found in Kay et al. (2015) and Yang et al. (2020), respectively.
The variables employed here comprise precipitation and wind in atmosphere component of Community Atmospheric
Model version 5 (CAM5), with a spatial resolution of 1.25 ° in longitude and 0.9 ° in latitude; and SST in the ocean



component of Parallel Ocean Program version 2 (POP), with 320 grids in longitude and 384 grids in latitude. Before
further analyses, model outputs are interpolated at a resolution of 2 °×2 °using a bilinear interpolation method (Mastyło,
2013), identical to that of 20CRv2c. In the current study, we focus on the historical simulation period of 1920–2005.

Here, it is important to stress the following two points. First, although the TIO domain in CESM1_IOPES is broader than
that for defining $I_{IOBM}$, there exist highly consistent temporal variations in SSTAs between them in observations (**Fig. S2**)
and simulations (**Fig. S3**) at interdecadal timescales, with temporal correlation coefficients (TCCs) of 0.93 and 0.87 ($P <$
0.01), respectively. Second, when selecting the SSTAs over the broader TIO domain (purple box in **Fig. S4**) as a metric,
it can be found that the observed (**Fig. S4a**) and modelled (**Fig. S4b**) large and intense loadings of the positive SSTAs are
still concentrated around the narrower TIO domain (black box in **Fig. S4**). As such, it is plausible to adopt the above-
mentioned Indian Ocean pacemaker experiment with broader TIO SSTAs to validate our proposed mechanisms tied to
the interdecadal IOBM variations.

## 3 Results



### 3.1 Observed interdecadal variations of the summer precipitation over EAMBZ and related background circulations




**Figure 1c** plots the spatial distribution of the interdecadal standard deviation of precipitation. This distribution is quite
similar to that of the climatology (**Fig. 1b**), suggesting relatively strong (weak) interdecadal precipitation fluctuations
over the EASM-dominated (westerly-controlled) part of the EAMBZ. Moreover, we show the first EOF mode of JJA-
mean EAMBZ precipitation (**Fig. 1d**), which accounts for 28% of the total variance and distinguishes from the remaining
eigenvectors according to the criterion defined by North et al. (1982). The leading EOF mode bears close resemblance to
the standard deviation of the EAMBZ precipitation on interdecadal timescales (**Figs. 1c** and **1d**), with larger loadings
occupying the Bohai Sea and Korean peninsula and their adjoining regions. The interdecadal TCC between the principal
component of the EOF1 and area-averaged precipitation over the research domain of EAMBZ (35 °–55 °N, 105 °–130 °E)
[EAMBZ precipitation index ($I_{EAMBZP}$); **Fig. 1e**] is 0.93 ($P < 0.001$). The aforementioned results indicate that that our
defined $I_{EAMBZP}$ can serve as a good indicator of the predominant fluctuations in the precipitation anomalies over EAMBZ
at interdecadal timescales. As such, from the time series of 11-year low-passed filtered $I_{EAMBZP}$ (**Fig. 1e**), we can observe
that the summer EAMBZ precipitation delineates notable interdecadal fluctuations. For example, EAMBZ experienced
dry summers during the periods preceding 1927, 1968–1982, and 1998–2010, but underwent wet summers during the
periods of 1928–1938, 1946–1967, and 2011 onwards. These notable interdecadal fluctuations are basically consistent
with those suggested by Si et al. (2021).

Before examining the modulation of IOBM on the interdecadal EAMBZ precipitation fluctuations, it is essential to
scrutinize the JJA-mean $I_{EAMBZP}$-associated circulation anomalies. The highest mid-latitude positive correlation region
can be discerned north of the TP (38 °–46 °N, 80 °–112.5 °E; blue box in **Fig. 2a**), suggesting that the interdecadal
enhancement of the summer EAMBZ precipitation is intimately correlated with the acceleration of the upstream mid-
latitude westerlies at 400 hPa. In light of the method of Chen et al. (2021) and J. Wang et al. (2022), we correlate the
$I_{EAMBZP}$ with the zonal winds averaged over the longitudinal range of EAMBZ at multiple levels (**Fig. 2b**) to further check
whether the most significant correlation occurs at 400 hPa. Evidently, on interdecadal timescales, the largest positive
correlation between precipitation and mid-latitude westerlies within 38 °–46 °N does occur at the mid-tropospheric level
of 400 hPa, with a TCC of 0.46 ($P < 0.01$) between the $I_{EAMBZP}$ and areal mean 400-hPa zonal wind (U400) over the



upstream westerly-dominated domain (**Fig. 2c**). Note that this correlation pattern exhibits a barotropic structure (**Fig. 2b**).
Additionally, we correlate the $I_{EAMBZP}$ with the 850-hPa meridional winds (V850). The $I_{EAMBZP}$ is positively correlated
with the key monsoonal southerly domain east of the TP (25 °–33 °N, 102.5 °–112.5 °E; green box in **Fig. 2d**), which is
located in the western portion of the EASM domain (Ying et al., 2023). The interdecadal correlation pattern between
meridional winds and the summer EAMBZ precipitation at multiple levels exhibits a baroclinic structure, with the
significant positive correlations confined below 500 hPa (**Fig. 2e**). Note that the strongest positive correlation is detected
at 850 hPa within 102.5 °–112.5 °E, with a TCC of 0.63 (**Fig. 2f**; $P < 0.001$) between $I_{EAMBZP}$ and areal mean V850 over
the key EASM-controlled domain (**Fig. 2d**).

**Figure 3** gives the JJA-mean $I_{EAMBZP}$-regressed circulation anomalies at interdecadal timescales. The interdecadal
enhancement of the EAMBZ precipitation is significantly linked to a localized quasi-barotropic cyclonic (low-pressure)
anomaly. At 400 hPa, significant westerly anomalies prevail in its southern flank, inducing the acceleration of westerlies
upstream of EAMBZ (**Fig. 3a**). At 850 hPa, the enhanced EAMBZ precipitation is connected to a north-south meridional
seesaw pattern, with a significant anticyclonic (high-pressure) anomaly over the subtropical western Pacific (SWP) and a
significant cyclonic anomaly over EAMBZ (**Fig. 3b**), exhibiting a somewhat barotropic structure (**Figs. 3a** and **3b**).
Significant southerly anomalies prevail in the western flank of this SWP clockwise gyre anomaly (SWPCGA). Moreover,
from the perspective of <WVT> (**Fig. 3c**), the magnitudes of southerly <WVT> anomalies over the key EASM-controlled
domain tied to the SWPCGA are much greater than the westerly <WVT> anomalies over the westerly-dominated domain.
Note that the southerly <WVT> anomalies are significantly divergent, pushing copious amounts of warm and moist vapor
over the SWP into EAMBZ. Then, with the aid of the local anticlockwise <WVT> gyre pattern (**Fig. 3c**), the EASM
southerlies from the low latitudes, which bring warm temperature advection anomalies, may easily collide with the mid-
level cold temperature advection anomalies brought by mid-latitude enhanced westerlies (**Figs. 4a** and **4b**), manifesting
the extratropical–tropical interplay around EAMBZ on interdecadal timescales. Such interplay is basically aligned with
that on interannual timescales (cf. Chen et al., 2021). Under such environments, atmospheric instability over EAMBZ can
be triggered to generate in situ significant ascending motion anomalies responsible for increased precipitation (**Fig. 5a**).
Note that considering the greater magnitudes of anomalies of <WVT> and warm temperature advection connected to the
southerlies over the key EASM-controlled domain, we presume that the monsoonal southerlies play a predominant
dynamical role in the interdecadal enhancement of precipitation over EAMBZ. To verify this presumption, we further
propose an East Asian monsoon index ($I_{MI}$), defined as the areal mean V850 over the key monsoonal southerly domain,
and a westerly index ($I_{WI}$), defined as the areal mean U400 over the upstream westerly-dominated region. The $I_{MI}$-
regressed results can well and realistically reproduce the magnitudes and distributions of the anomalous upward motions
tied to $I_{EAMBZP}$ (**Fig. 5b** vs. **5a**). However, the magnitudes of $I_{WI}$-regressed results are highly weakened, along with the
major loadings shifting more southward (**Fig. 5c**). Above results could allow us to conclude that the anomalous southerlies
over the key monsoonal southerly domain could be the predominant driving factor for the interdecadal enhancement of
summer EAMBZ precipitation, whereas the upstream accelerated westerlies play a secondary dynamical amplification
role.

**3.2 Interdecadal relationship between IOBM and the summer EAMBZ precipitation**

Many previous studies have substantiated that the IOBM can remotely modulate summer rainfall fluctuations over the
mid-latitude Asia at interdecadal timescales (e.g., Zhang et al., 2018; S. Wang et al., 2022; Wu et al., 2022). In the present
study, we also identify that the IOBM can exert profound impacts on the interdecadal variations of the EAMBZ
precipitation in boreal summer. In this subsection, we firstly reveal their intimate relationship. **Figure 6a** exhibits the





correlation pattern between the JJA-mean $I_{EAMBZP}$ and the contemporaneous global gridded SST at interdecadal timescales.
The most pronounced and significant correlations are found in the TIO sector, which largely matches the domain for
delineating the IOBM mode (black frame in **Fig. 6a**). There exists a salient out-of-phase relationship between the
interdecadal EAMBZ precipitation changes and the IOBM mode, with a TCC of –0.57 between $I_{EAMBZP}$ and $I_{IOBM}$ (**Fig.**
**6b**; $P < 0.01$). This result suggests that IOBM warming (cooling) is significantly connected with dry (wet) EAMBZ
summers, which serves as a critical oceanic modulator. On interdecadal timescales, the IOBM can remotely spark
conducive dynamical circumstances for increased precipitation over EAMBZ, i.e., the collision between cold and warm
airflows around EAMBZ (**Figs. 4c** and **4d**) and the locally significant convergent ascending motion anomalies resembled
those tied to the positive $I_{EAMBZP}$ (**Fig. 5d** vs. **5a**). However, the extratropical cold (tropical warm) temperature advection
anomalies west (south) of the EAMBZ, which are tied to the strengthened westerlies (southerlies), are quite insignificant
(significant) (**Figs. 4c** and **4d**). This indicates that the IOBM may exerts a more profound influence on the southerly wind
anomalies over the EASM-controlled domain, which is more important for enhanced EAMBZ precipitation; whereas the
IOBM may insignificantly modulate the westerly anomalies over the westerly-dominated region. The possible underlying
mechanisms of how IOBM links the summertime circulation anomalies responsible for the interdecadal fluctuations in
the EAMBZ precipitation will be illuminated in the next subsection.

**3.3 Possible mechanisms**

**Figure 7** shows partial regression of the JJA-mean anomalies of SST and large-scale precipitation over TIO and its
neighboring areas onto the $I_{IOBM}$ at interdecadal timescales with the IPO signal removed. Corresponding to higher $I_{IOBM}$
years, warm SSTAs cover almost all areas of TIO, with large loadings appearing in the central-southern TIO and relatively
small loadings appearing in the northern TIO (**Fig. 7a**), which are aligned with the previous studies (Wu et al., 2016; Y.
Huang et al., 2019). Moreover, there are striking suppressed precipitation around the northeast corner of the TIO domain
(**Fig. 7b**), suggesting profoundly localized atmospheric responses to the warm TIO SSTAs. Note that there exist positive
precipitation anomalies around the northeast corner of TIO during the cold TIO SST years, suggesting the release of
anomalous atmospheric heating. Since the significant out-of-phase relationship between summertime IOBM and EAMBZ
precipitation at interdecadal timescales, we adopt negative $I_{IOBM}$-regressed patterns to express the influence of cold SSTAs
over the TIO region. **Figure 8** displays the anomalous patterns of the RWS, velocity potential, and divergent horizontal
winds regressed onto the negative $I_{IOBM}$. The velocity potential anomalies with larger negative (positive) loadings in the
upper (lower) troposphere are concentrated surrounding the northeast corner of TIO. Under these circumstances, local
upper (lower) tropospheric divergence (convergence) and negative (positive) RWS anomalies can be observed (**Fig. 8**),
suggesting enhanced ascending motions and convection activities in situ and thereby exciting the localized increased
precipitation/atmospheric heating. The above results indicate that IOBM cooling may transmit its interdecadal influence
via the intermediate atmospheric bridge of enhanced convective activities around the northeast corner of TIO, exerting a
remote modulation on the interdecadal EAMBZ rainfall variations.

Next, we further discuss the physical pathway linking IOBM cooling with the far-reaching downstream circulation
anomalies responsible for the interdecadal enhancement of EAMBZ precipitation, as shown in **Fig. 9**. Because the
cyclonic anomaly at 400 hPa shifts more eastward compared to the $I_{EAMBZP}$-regressed counterpart (**Fig. 9a** vs. **3a**), only
fractional westerly anomalies occupy the eastern part of the westerly-dominated region. The TCC between $I_{IOBM}$ and $I_{WI}$
is nearly equal to zero ($r = –0.06$), thus linking the insignificant cold temperature advection displayed in **Fig. 4c**.
Nevertheless, in the lower troposphere, a "north-low–south-high" meridional seesaw pattern over the Northeast China–
SWP sector is found to be linked with IOBM cooling (**Fig. 9b**). Note that this negative $I_{IOBM}$-regressed seesaw pattern





exhibits a quasi-barotropic structure, with an anticlockwise <WVT> gyre in the north and a SWPCGA in the south (**Fig. 9c**), which is highly similar to that shown in **Fig. 3**. Significant anomalies of V850 and southerly <WVT> prevail over the key monsoonal southerly domain, lying on the western flank of SWPCGA (**Figs. 9b** and **c**). The TCC between $I_{IOBM}$ and $I_{MI}$ is –0.33, significant at 0.05 on interdecadal timescales, thereby linking the significant warm temperature advection anomalies indicated in **Fig. 4d**.

One may ask how IOBM cooling induces the above-mentioned meridional seesaw pattern. Previously, we have revealed that negative SSTAs over TIO may exert remote interdecadal impacts through an atmospheric bridge, i.e., vigorous convective activities around the northeast corner of TIO (**Figs. 7** and **8**). In effect, there exists a low-level cyclonic anomaly in situ (**Fig. 9b**). Such cyclonic anomaly can be interpreted as a typical Gill–Matsuno-type response to regional atmospheric heating caused by IOBM cooling (Matsuno, 1966; Gill, 1980), which is more clear within the lower levels (**Fig. 9b**). As a result, consistent easterly anomalies appear from SWP to its northern flank around 15 °N, denoting the active role of depressed air pressure. The consistent easterly anomalies over SWP could lead to local anticyclonic wind shear anomalies (Wang et al., 2019). In such a scenario, a quasi-barotropic SWPCGA can be induced (**Fig. 9**). Further, local downward motions tied to SWPCGA could induce significant upward motions to its north via a meridional overturning circulation (J. Wang et al., 2021), thus exciting a quasi-barotropic cyclonic anomaly and an anticlockwise <WVT> gyre pattern centered over Northeast China (**Fig. 9**). Therefore, positive summertime rainfall anomalies over EAMBZ at interdecadal timescales can be induced.

## 3.4 Estimation of the interdecadal variations of summer EAMBZ precipitation

In the last two subsections, we suggest that the IOBM cooling can serve as a significant oceanic modulator for increased summer EAMBZ precipitation at interdecadal timescales, and present the corresponding physical mechanisms. To estimate their steady antiphase relationship, in the following, the negative $I_{IOBM}$ is selected to construct a physical-based empirical (P-E) model by using the simple linear regression (SLR) analysis and the cross-validation method (You and Jia, 2018; Chang et al., 2021; Jeong et al., 2021), representing the impact of IOBM cooling. The P-E model is given as follows:

$$I_{EAMBZP} = \beta_0 + \beta_1 I_{IOBM} + \varepsilon, \quad (5)$$

where $\beta_0$ and $\beta_1$ are regression coefficients, and $\varepsilon$ denotes the residuals. The time series of $I_{EAMBZP}$ and $I_{IOBM}$ are detrended and 11-year low-pass filtered beforehand.

Following Jeong et al. (2021), a "leaving one out" cross-validation strategy is employed to determine the robustness of the hindcast estimates. The normalized time series of summer $I_{EAMBZP}$ and associated leave-one-out cross-validated hindcast estimates are shown in **Fig. 10**. The TCC between P-E predicted hindcast estimates (blue line) and the observed $I_{EAMBZP}$ (red line) for 1901–2014 can reach 0.56 ($P < 0.05$), suggesting that the P-E model can well capture the interdecadal $I_{EAMBZP}$ variations and reflect their steady relationship.

## 4 Discussion

Numerous recent studies employed the pacemaker experiments that restore the historical observational basin-wide SSTAs to validate the mechanisms of how remote SSTAs over the Pacific Ocean, the Atlantic Ocean, and the Indian Ocean modulate the climate anomalies in many areas across the globe at interdecadal timescale (e.g., Yang et al., 2020; Jiang et



al., 2021; J. Wang et al., 2022). As proposed in Section 2.6, there exists a considerably high positive TCC between the
broader TIO SSTAs regarding CESM1_IOPES and narrower TIO SSTAs regarding IOBM in both observations and
simulations. Furthermore, the simulated large and strong loadings of SSTAs concentrate over the central-southern TIO
(**Fig. S4b**), delineating a quite similar distribution with that in the observation (**Figs. 7a** and **S4a**). As a consequence, we
use the pacemaker experimental data based on the ensemble mean of CESM1_IOPES and CESM1_LENS to validate our
proposed mechanisms regarding the modulation of IOBM cooling on the interdecadal enhancement of summer EAMBZ
precipitation. Considering the predominant role of southerly anomalies over the key monsoonal southerly domain, we
therefore emphasize the low-level (850 hPa) atmospheric anomalies at interdecadal timescales tied to the IOBM-like SST
cooling, as depicted in **Fig. 11**. We can observe a clearly anomalous cyclonic circulation around the northeast corner of
TIO, accompanied by local positive precipitation anomalies and easterly anomalies that stretch from SWP to its northern
flank, which are generally resembled those in the observation (**Figs. 7b** and **9b**). In this circumstance, a similar "north-
low–south-high" meridional seesaw pattern over the Northeast China–SWP sector can be formed to spark and sustain the
enhanced EAMBZ precipitation in boreal summer (**Fig. 11**). However, although the results from CESM1_LENS and
CESM1_IOPES can reasonably confirm our proposed physical pathway of how IOBM cooling exerts a distant modulation
on the interdecadal enhancement of summer precipitation over EAMBZ, we can still notice the weakness of the model
simulations. That is, positive precipitation anomalies around the northeast corner of TIO and the easterly anomalies
exhibit weaker magnitudes compared to the observations (**Fig. 11** vs. **7b** and **9b**). Besides, systematic biases exist
regarding the simulated positions of the upper (lower) tropospheric divergence (convergence) and negative (positive)
RWS anomalies (**Fig. S5**), manifesting themselves in the eastward displacement tendency in contrast to those around the
northeast corner of the TIO (**Fig. 8**).

## 5 Conclusions

In this study, by analysis of the long-term observational and reanalysis datasets during 1901–2014, the temporal
characteristics of interdecadal variations in the summer EAMBZ precipitation and associated circulation background are
revealed. The potential modulation of IOBM on the variations is further discussed.
The summer EAMBZ precipitation exhibited a salient interdecadal fluctuations, e.g., with dry summers during the periods
preceding 1927, 1968–1982, and 1998–2010, as well as wet summers during the periods of 1928–1938, 1946–1967, and
2011 onwards. It is indicated that the cold airflows brought by the mid-latitude accelerated upstream westerlies over the
westerly-dominated domain collide and converge with the warm and humid airflows brought by the enhanced southerlies
over the key EASM-controlled domain, suggesting the local extratropical–tropical interplay. Further diagnostic results
suggest that the monsoonal southerly anomalies could be viewed as the predominant driving factor for the interdecadal
enhancement of EAMBZ precipitation, whereas the upstream westerlies play a secondary dynamical amplification role.
Such circulation anomalies are closely linked to a "north-low–south-high" meridional seesaw pattern over the Northeast
China–SWP sector, which provides favorable environments for the transportation of water vapor from the SWP and the
convergence over EAMBZ to spark enhanced summer EAMBZ precipitation at interdecadal timescales.
We further identify that the IOBM-related SST anomaly pattern is a salient oceanic modulator for the interdecadal
variations of the summer EAMBZ precipitation via the Gill–Matsuno mechanism. When the cold phase of the IOBM
occurs, an anomalous cyclonic circulation is excited around the northeast corner of TIO. As a response, consistent easterly
anomalies appear from SWP to its northern flank, leading to local anticyclonic wind shear anomalies and thus inducing a
SWPCGA pattern and a resultant anticlockwise gyre pattern centered over Northeast China. On interdecadal timescales,



such meridional seesaw pattern tied to the IOBM cooling is responsible for enhanced summer precipitation over EAMBZ
through linking the predominant driving factor of strengthened monsoonal southerly anomalies west of the SWPCGA
pattern. As such, the water vapor transportation from the SWP and the convergence over EAMBZ can be triggered to
induce and sustain the enhancement local precipitation. Correspondingly, a P-E model based the negative $I_{IOBM}$ is
constructed, which can well capture the interdecadal fluctuations in the EAMBZ precipitation and reflect their steady
relationship. Furthermore, the results based on the large ensemble experiment and the Indian Ocean pacemaker
experiment also confirm the crucial physical pathway linking the SST variations over TIO with the summer precipitation
over EAMBZ via the influence of SST variations on the aforementioned meridional seesaw pattern at interdecadal
timescales.
**Code and data availability.** The CRU TS precipitation data version 3.26 (CRU TS3.26) from CRU at the University of
East Anglia are available online (https://catalogue.ceda.ac.uk/uuid/3f8944800cc48e1cbc29a5ee12d8542d; CRU, 2022).
The ERSSTv5 data from the US NOAA are available from the following website:
https://www1.ncdc.noaa.gov/pub/data/cmb/ersst/v5/netcdf/ (NOAA 2020). The 20CRv2c datasets from NOAA-CIRES
are available from the following website: https://psl.noaa.gov/data/gridded/data.20thC_ReanV2c.html (NOAA-CIRES,
2022). The model simulation datasets regarding CESM1_LENS are available online
(https://www.cesm.ucar.edu/community-projects/lens/data-sets; NCAR 2023). The model simulation datasets regarding
CESM1_IOPES are available online (https://www.earthsystemgrid.org/dataset/ucar.cgd.ccsm4.IOD-PACEMAKER.html;
NCAR 2023).
Codes are available from the corresponding author on reasonable request.
**Author contributions.** YL designed the research; JW wrote the first draft of the paper; FC and CS downloaded and
analyzed the data, and plotted the figures used in this study. All authors, including YD and XX, contributed to the
discussion of the results and reviewed the manuscript.
**Competing Interests.** The contact author has declared that none of the authors has any competing interests.
**Acknowledgements.** Yanju Liu acknowledges the support by the Key Innovation Team of China Meteorological
Administration "Climate Change Detection and Response" (CMA2022ZD03).
**Financial support.** This study was supported by the Second Tibetan Plateau Scientific Expedition and Research (STEP)
program (2019QZKK010204-02 and 2019QZKK0102), Guangdong Major Project of Basic and Applied Basic Research
(2020B0301030004), and Innovation and Development Special Project of China Meteorological Administration
(CXFZ2022J039).

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

**Figures**

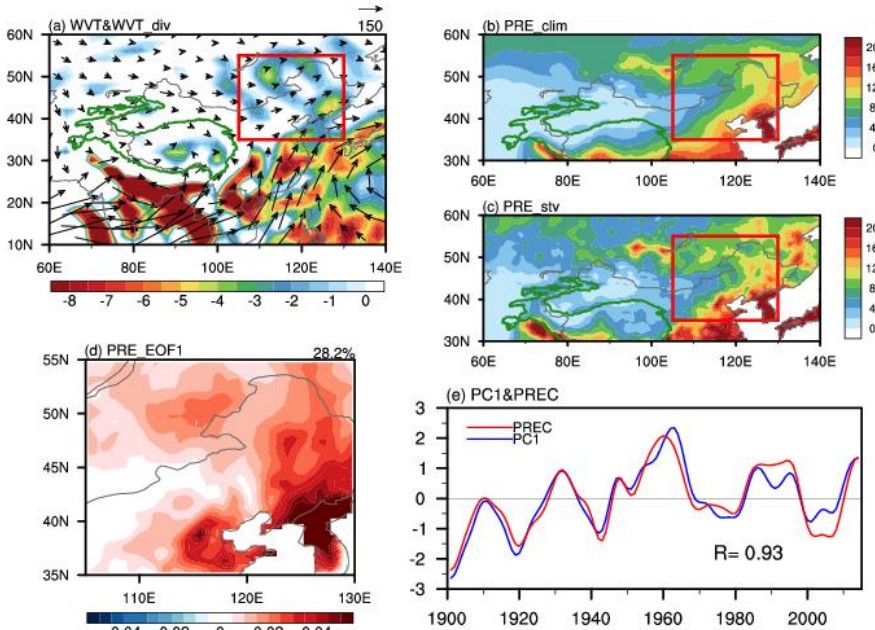

**Figure 1.** The climatological JJA-averaged (a) <WVT> (vectors; kg m$^{-1}$ s$^{-1}$) and <WVT_div> (shading; 10$^{-5}$ kg m$^{-2}$ s$^{-1}$), (b) precipitation (mm month$^{-1}$), and (c) interdecadal standard deviation of precipitation (mm month$^{-1}$) during the period 1901–2014. The red box (35 °–55 °N, 105 °–130 °E) outlines the research domain of EAMBZ (the same hereinafter). (d) Spatial pattern of the first empirical orthogonal function (EOF1) mode of JJA-mean EAMBZ precipitation. (e) Normalized time series of the JJA-mean EAMBZ precipitation index ($I_{EAMBZP}$) (red line) and associated first principal component (PC1) (blue line), with the number denoting the temporal correlation coefficient (TCC) between the corresponding time series. In panels (c)–(e), variables are detrended and 11-year low-pass filtered. The green outline in panels (a)–(c) represents the terrain of the Tibetan Plateau (TP) at 2000 m (the same hereinafter). The precipitation is derived from the CRU TS3.26 precipitation data, while other variables are from the 20CRv2c datasets.

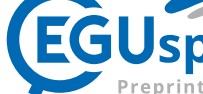

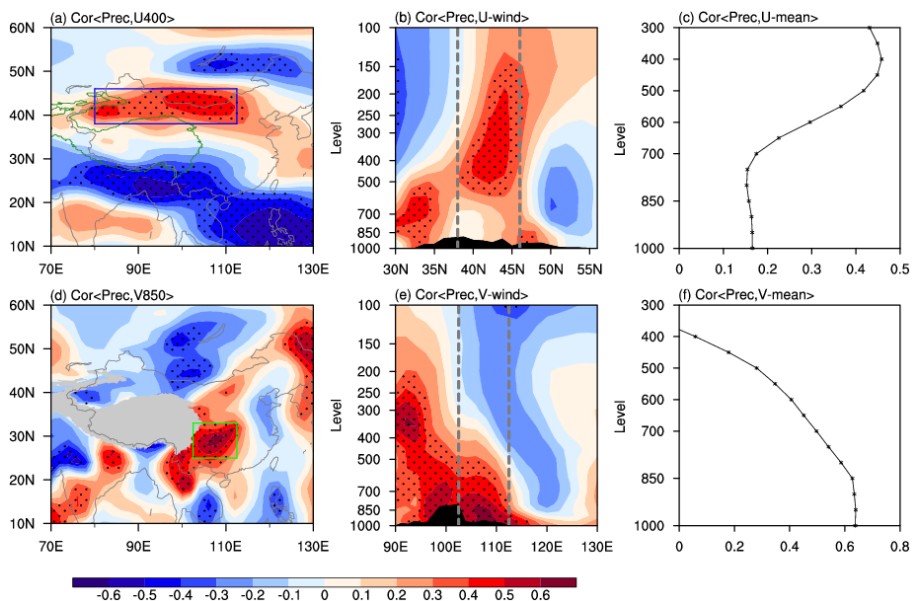

**Figure 2.** Correlation maps of the JJA-averaged $I_{EAMBZP}$ with the simultaneous (a) 400-hPa zonal wind (U400) and (d) 850-hPa meridional wind (V850), and (b) height–latitude cross-section of zonal winds averaged over 80 °–112.5 °E, and (e) height–longitude cross-section of meridional winds averaged over 25 °–33 °N, during the period 1901–2014. The blue box (38 °–46 °N, 80 °–112.5 °E) in (a) and the green box (25 °–33 °N, 102.5 °–112.5 °E) in (d) represent the upstream westerly domain and the monsoonal southerly domain significantly tied to the interdecadal variations of precipitation over EAMBZ, respectively (the same hereinafter). The grey-dashed vertical lines in (b) and (e) represent the latitudinal and longitudinal range of the westerly and the monsoonal southerly domain, respectively. (c) Profile of correlation coefficients between the JJA-averaged $I_{EAMBZP}$ and the simultaneous area-averaged zonal winds over the upstream westerly domain at multiple levels during the period 1901–2014. (f) As in (c), but for the meridional winds over the monsoonal southerly domain. All variables are detrended and 11-year low-pass filtered. Areas with significant values exceeding the 95% confidence level are stippled. The black shading indicates the topography. The grey shaded areas denote the TP areas above 2000 m (the same hereinafter). The $I_{EAMBZP}$ is calculated based on the CRU TS3.26 precipitation data, while other variables are from the 20CRv2c datasets.

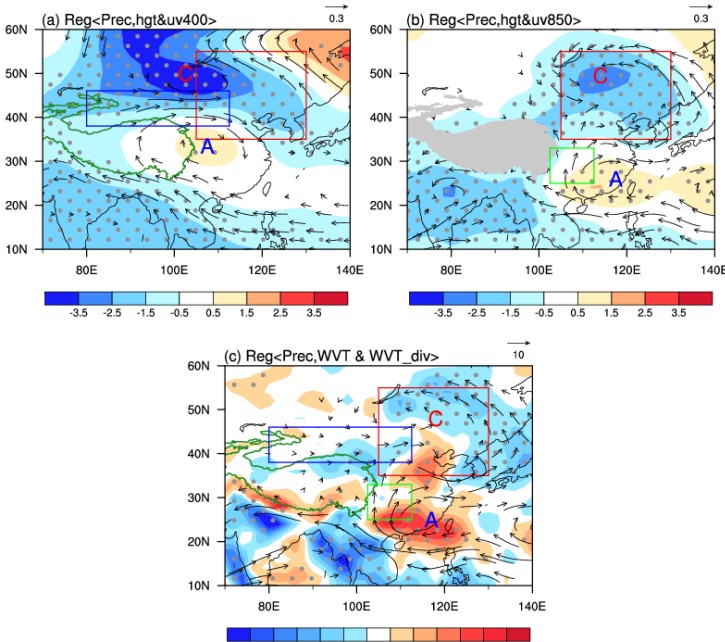

**Figure 3.** Regression maps of the JJA-mean anomalies of (a) 400-hPa geopotential height (Z400; shading; m) and wind field (UV400; vectors; m s$^{-1}$), (b) 850-hPa geopotential height (Z850; shading; m) and wind field (UV850; vectors; m s$^{-1}$), and (c) <WVT> (vectors; kg m$^{-1}$ s$^{-1}$) and <WVT_div> (shading; 10$^{-5}$ kg m$^{-2}$ s$^{-1}$) onto the concurrent $I_{EAMBZP}$ during the period 1901–2014. All variables are detrended and 11-year low-pass filtered. Letter A (C) represents the center of anticyclonic (cyclonic) anomaly (the same hereinafter). Areas with significant values of Z400, Z850, and <WVT_div> that exceed the 95% confidence level are stippled, respectively. Only vectors that are significant at the 95% confidence level are shown. The $I_{EAMBZP}$ is calculated based on the CRU TS3.26 precipitation data, while other variables are from the 20CRv2c datasets.

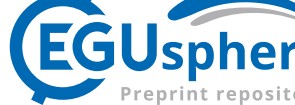

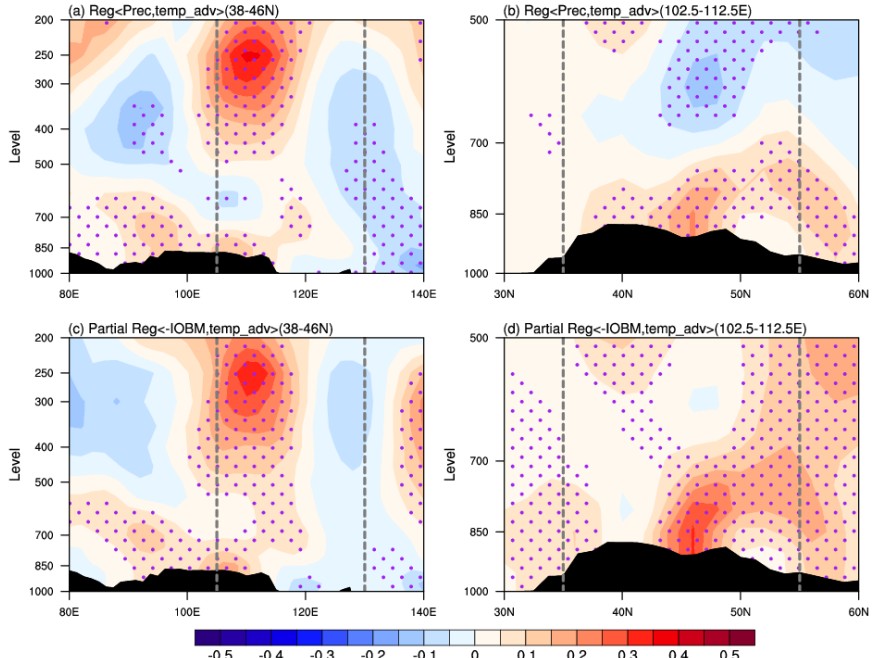

**Figure 4.** (a) Height–longitude cross-section (averaged over 38 °–46 °N) and (b) height–latitude cross-section (averaged over 102.5 °– 112.5 °E) of the JJA-mean temperature advection anomalies (shading; $10^{-5}$ K s$^{-1}$) regressed onto the concurrent $I_{EAMBZP}$ during the period 1901–2014. (c, d) As in (a, b), but for patterns of the partial regression coefficient between temperature advection and negative $I_{IOBM}$ without the IPO signal. The gray vertical lines in (a, c) and (b, d) represent the longitudinal and latitudinal range of the research domain of EAMBZ, respectively. The black shading indicates the topography. All variables are detrended and 11-year low-pass filtered. Areas with significant values exceeding the 95% confidence level are stippled. The $I_{EAMBZP}$ and $I_{IOBM}$/IPO index are calculated based on the CRU TS3.26 precipitation data and the ERSSTv5 dataset, respectively; whilst other variables are from the 20CRv2c datasets.



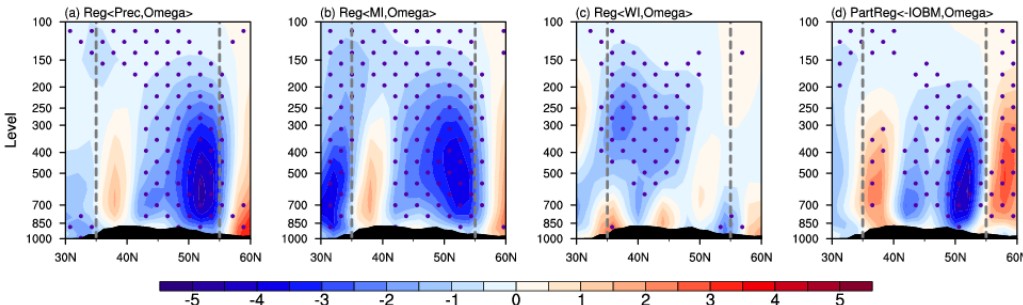

**Figure 5.** Height–latitude cross-section (averaged over 105 °–130 °E) of the JJA-mean vertical velocity anomalies ($10^{-3}\,\text{Pa s}^{-1}$) regressed onto the concurrent (a) $I_{\text{EAMBZP}}$, (b) $I_{\text{MI}}$, and (c) $I_{\text{WI}}$ during the period 1901–2014. (d) As in (a), but for the partial regressed anomalies onto the negative $I_{\text{IOBM}}$ with the IPO signal removed. The gray vertical lines represent the latitudinal range of EAMBZ. The black shading indicates the topography. All variables are detrended and 11-year low-pass filtered. Areas with significant values exceeding the 95% confidence level are stippled. The $I_{\text{EAMBZP}}$ and $I_{\text{IOBM}}$/IPO index are calculated based on the CRU TS3.26 precipitation data and the ERSSTv5 dataset, respectively; whilst other variables are from the 20CRv2c datasets.

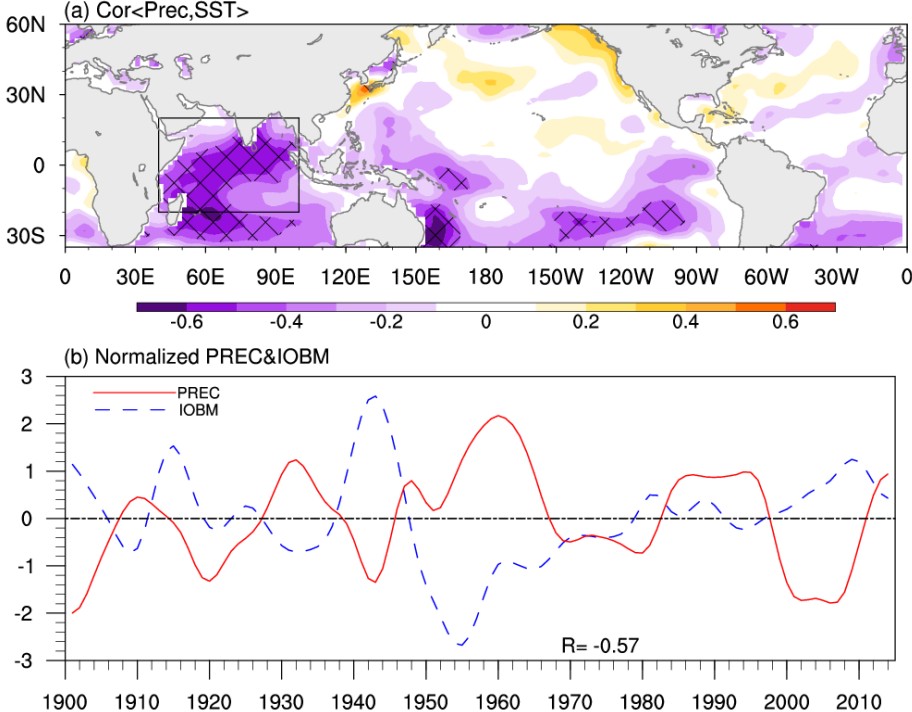

**Figure 6.** (a) Correlation map of the JJA-mean $I_{EAMBZP}$ with the concurrent near-global SST (35 °S–60 °N) during the period 1901–2014. The black frame (20 °S–20 °N, 40 °–100 °E) outlines the domain for delineating the IOBM mode (the same hereinafter). Areas with significant values exceeding the 99% confidence level are hatched. (b) Normalized time series of the JJA-mean $I_{EAMBZP}$ (red line) and $I_{IOBM}$ (blue line) from 1901 to 2014. The numeral at the bottom represents the TCC between the corresponding time series. All variables are detrended and 11-year low-pass filtered. The SST is from the ERSSTv5 dataset. The $I_{EAMBZP}$ and $I_{IOBM}$ are calculated based on the CRU TS3.26 precipitation data and the ERSSTv5 datasets, respectively.



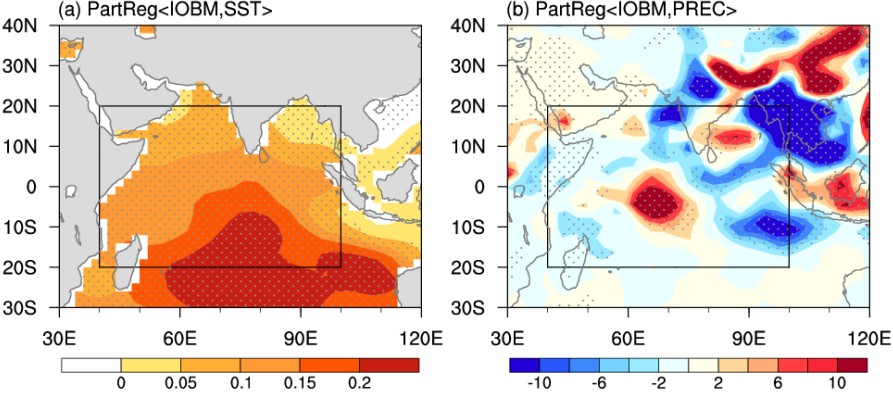

**Figure 7.** Partial regression of the JJA-mean (a) SST (°C) and (b) precipitation (mm month$^{-1}$) anomalies over TIO and its neighboring areas onto the concurrent $I_{IOBM}$ with the IPO signal removed for the period 1901–2014. All variables are detrended and 11-year low-pass filtered. Areas with significant values exceeding the 95% confidence level are stippled. The $I_{IOBM}$/IPO index is calculated based on the ERSSTv5 dataset. The SST and the precipitation are derived from the ERSSTv5 dataset and the 20CRv2c dataset, respectively.



821

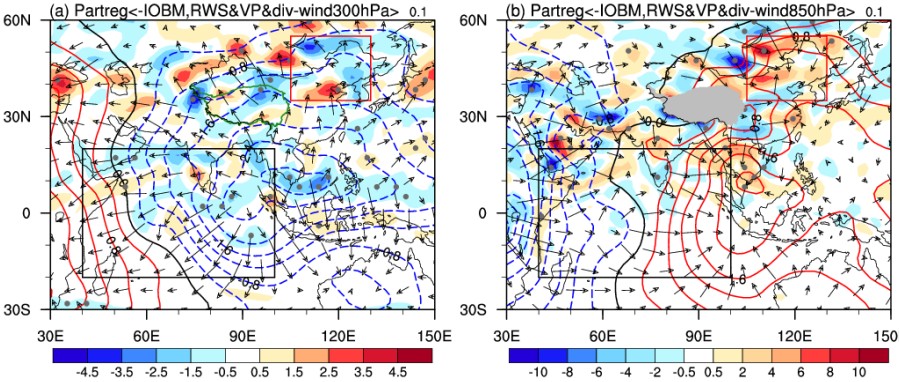

**Figure 8.** Partial regression of the JJA-mean (a) 300- and (b) 850-hPa RWS (shading; $10^{-11}\,\mathrm{s}^{-2}$), velocity potential (contours; interval: 0.4; $10^5\,\mathrm{m}^2\,\mathrm{s}^{-1}$), and divergent horizontal wind (vectors; $\mathrm{m\,s}^{-1}$) anomalies against the concurrent negative $I_{\mathrm{IOBM}}$ with the IPO signal removed during the period 1901–2014. All variables are detrended and 11-year low-pass filtered. Areas with significant values of RWS exceeding the 95% confidence level are stippled. The $I_{\mathrm{IOBM}}$/IPO index is calculated based on the ERSSTv5 dataset; whilst other variables are from the 20CRv2c datasets.






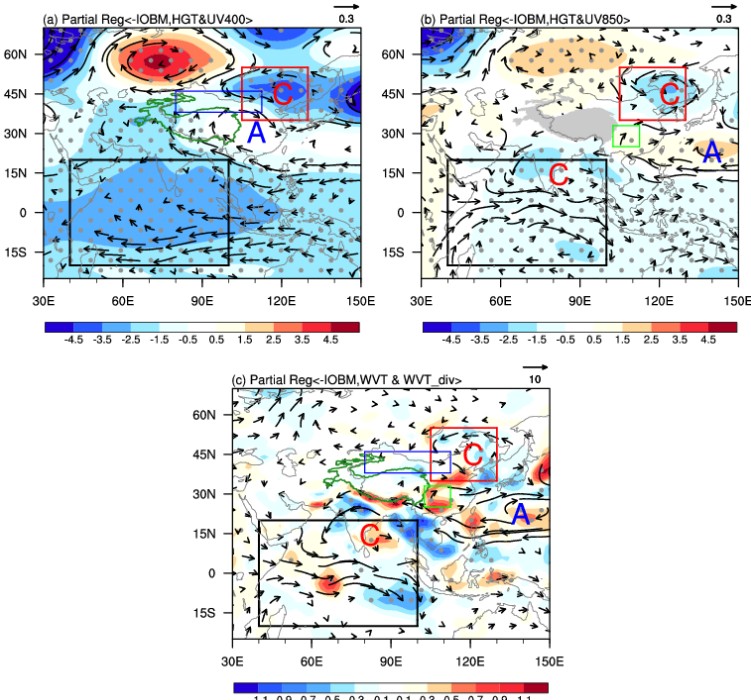

**Figure 9.** Partial regression of the JJA-mean (a) Z400 (shading; m) and UV400 (vectors; m s$^{-1}$), (b) Z850 (shading; m) and UV850 (vectors; m s$^{-1}$), and (c) <WVT> (vectors; kg m$^{-1}$ s$^{-1}$) and <WVT_div> (shading; $10^{-5}$ kg m$^{-2}$ s$^{-1}$) onto the concurrent negative $I_{IOBM}$ with the IPO signal removed during the period 1901–2014. All variables are detrended and 11-year low-pass filtered. Areas with significant values of Z400, Z850, and <WVT_div> that exceed the 95% confidence level are stippled, respectively. Only vectors that are significant at the 95% confidence level are shown. The $I_{IOBM}$/IPO index is calculated based on the ERSSTv5 dataset; whilst other variables are from the 20CRv2c datasets.



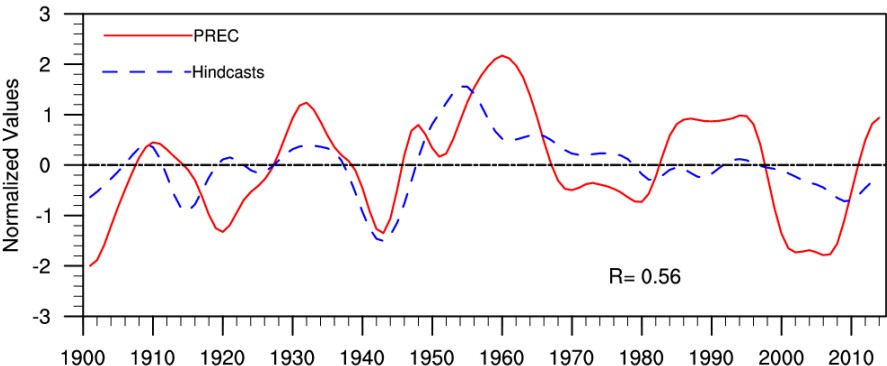

**Figure 10.** Normalized time series of the JJA-mean $I_{EAMBZP}$ (red line) and associated leave-one-out cross-validated hindcast estimates (blue line) for 1901–2014, with the number denoting the TCC between the corresponding time series.



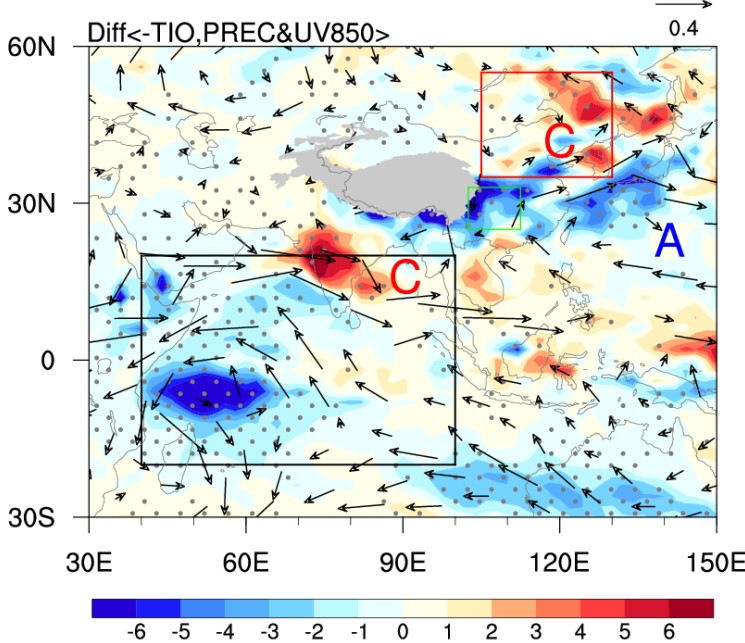

**Figure 11.** Simulated composite differences of JJA-mean UV850 (vectors; m s$^{-1}$) and precipitation (shading; mm month$^{-1}$) between cold and warm SST years over the broader TIO domain in CESM1_IOPES (15 °S–15 °N, 40 °–174 °E; purple box in **Fig. S4**). The warm and cold TIO SST years are selected based on the ±0.5 standard deviations of the simulated time-evolving SSTAs during 1920–2005, as shown in **Fig. S3** (red line). All variables are detrended and 11-year low-pass filtered. Areas with significant values of precipitation that exceed the 95% confidence level are stippled. Only vectors that are significant at the 95% confidence level are shown. The simulated anomalies of UV850 and precipitation are calculated based on the difference between the CESM1_IOPES ensemble mean and the CESM1_LENS ensemble mean (former minus latter), highlighting the internally driven impacts of TIO SSTAs.