# Peer review of "Role of Indian Ocean basin mode in driving the interdecadal variations of summer precipitation over the East Asian monsoon boundary zone"

_EGUsphere, 2023_

## Author Response (AR1)

This paper uses observations, re-analysis output and model simulations to examine interdecadal variability in the East Asian monsoon boundary zone, particularly in precipitation. The authors find that the cold phase of the Indian Ocean basin mode prompts anomalous cyclonic circulation over the north-eastern Indian Ocean, which ultimately enhances moisture transport from the Pacific Ocean to the boundary zone. I apologise to the authors for the delay in submitting this review.

Many thanks for your constructive and valuable comments, which have greatly improved our manuscript.

I have a few a few comments about this paper, primarily about the presentation and discussion. I recommend major revisions.

We have revised the manuscript based on your comments. The revisions are highlighted in red color in the revised manuscript. In the following, we summarize our point-by-point replies to your comments.

This paper uses far too many acronyms: I counted 30 in total. While some are fine to keep – if they are mentioned more than five or so times – others are used sparingly. Unfortunately, this makes large sections of the paper very difficult to follow: I spent a great deal of time trying to remember what each acronym was, or flicking backwards to look it up again. Consequently, I found the science message was often unclear. (And as something of an aside, a couple of acronyms were poorly chosen: EU commonly means European Union, and P-E could be confused with "precipitation minus evaporation", particularly to an audience of atmospheric scientists.)

**Reply:** We have removed the sparingly used acronyms (e.g., NH, AWJ, WNP, and ENSO) and kept the frequently used acronyms in the revised manuscript. For easy reading and reviewing, we have included the "Glossary of acronyms" in the Supplementary File. Please see the Glossary of acronyms in the Supplement File.

Furthermore, as you proposed, a couple of acronyms (i.e., EU and P-E) were poorly chosen owing to unclear science message conveying. Therefore, we abandoned these acronyms in the revised version.

The coastlines plotted on the figures are very faint: it is difficult to pick out the important features when it's unclear where they are. Also, axes and colour bars should be labelled on the figures as well as in the captions. And in Figure 6, hatching is used to indicate significance, whereas in other figures the authors use dots: please use just one for consistency.

**Reply:** We have modified associated figures for more conspicuous coastlines. For the consistency, we abandoned the hatching in **Figure 6**. Throughout the revised manuscript, we use grey dots to indicate significance. Please see the modified figures

in the revised version.

Moreover, you mentioned that axes and colour bars should be labelled on the figures as well as in the captions. After checking the papers regarding the climate dynamics that published in ACP, we found that the layout of our figures is quite consistent with those papers. For example, we scrutinized the axes and colour bars in **Figure 3** in the Paper "Yu, L., Zhong, S., Vihma, T., Sui, C., and Sun, B.: A change in the relation between the Subtropical Indian Ocean Dipole and the South Atlantic Ocean Dipole indices in the past four decades, Atmos. Chem. Phys., 23, 345–353, https://doi.org/10.5194/acp-23-345-2023, 2023".

We thus think that the layout (i.e., axes and colour bars) of our figures could be suitable.

[Figure]

**Figure 3** Regression maps of the SST anomalies (◦C) on the summertime indices of (a, b) SAOD and (c, d) SIOD over the periods of (a, c) 1979–1999 and (b, d) 2000–2020. Dots denote the regions where the confidence level is above 95 %. (from Yu et al., 2023, ACP)

I think the authors need to include more details about the model simulations described in Section 2.6 They subtract one set of simulations from the other, but it is not clear to me how this achieves the authors' stated goal (line 205). Please. Explain make clear how the two sets of simulations are different: why is internal variability arising from Indian Ocean SSTs unique to one simulation and not the other? And indeed – and I apologise if I have missed something – it is unclear how these models are used subsequently. I think it would be helpful to note, as the results are discussed, which data sets are being used at each point.

**Reply:** Thanks for your constructive comments and queries. Please see **Line 213-219** for the answers. More details about CESM1_LENS and CESM1_IOPES can be found in Kay et al. (2015) and Yang et al. (2020), respectively.

**Line 213-219**

"As indicated by Yang et al. (2020), the CESM1_LENS 35-member ensemble mean results can better provide an estimate of the influence of the external radiative forcing signals (e.g., greenhouse gas) on the climate system. Furthermore, the 10-member ensemble mean results in CESM1_IOPES contain the responses to both external forcings and the observed SST variations over the TIO domain (Yang et al., 2020). Therefore, by subtracting the CESM1_LENS ensemble mean from the

CESM1_IOPES ensemble mean, we can obtain responses of the climate system to the internal variability stemming from the time-varying TIO SSTAs, distinguishing the impact of external radiative force changes from the intrinsic variability driven by TIO SSTAs."

**Reference:**

Kay, J.E., Deser, C., Phillips, A., Mai, A., Hannay, C., Strand, G., Arblaster, J.M., Bates, S.C., Danabasoglu, G., Edwards, J., Holland, M., Kushner, P., Lamarque, J.F., Lawrence, D., Lindsay, K., Middleton, A., Munoz, E., Neale, R., Oleson, K., Polvani, L. and Vertenstein, M., 2015. The community earth system model (CESM) large ensemble project: a community resource for studying climate change in the presence of internal climate variability. Bulletin of the American Meteorological Society, 96(8): 1333-1349.

Yang, D., Arblaster, J.M., Meehl, G.A., England, M.H., Lim, E.-P., Bates, S. and Rosenbloom, N., 2020. Role of tropical variability in driving decadal shifts in the Southern Hemisphere summertime eddy-driven jet. Journal of Climate, 33(13): 5445-5463.

Furthermore, we used the model simulations (see the subsequent **Section 3.4**) to validate our proposed mechanisms of how the remote IOBM modulates the summertime EAMBZ precipitation at interdecadal timescales, aiming at providing more confident results. In other words, the statistical result has to be compared against the numerical model result. If they are consistent, we could confidently indicate that our proposed mechanisms are reliable.

The physical-based empirical model (Section 3.4) appears to be behind a key result of the paper: it is mentioned in the abstract. But this section feels rather brief. Could the authors perhaps discuss the implications of their model a bit further? They say it captures some of the observed interdecadal variability: what about its shortcomings? How is this result helpful?

**Reply:** Thanks for your insightful comments. We have extended the discussion concerning the shortcomings and the helpful aspect of our proposed physical-based empirical model in Section 3.5. Please see **Line 411-416** and **Line 461-466** in the revised version.

**Line 411-416**

"Although our proposed physical-based empirical model could confirm the concurrently intimately interdecadal relationship between IOBM and EAMBZ precipitation, we should acknowledge the shortcomings of the model. First, the amplitudes of the hindcast estimates are fairly lower, which cannot well capture the extreme precipitation years (e.g., years around 1960; Fig. 11). Second, the simultaneous signal of IOBM cannot be served as a predictor for summertime EAMBZ precipitation variations. As such, this model inherently lacks the ability to predict the interdecadal EAMBZ precipitation anomalies in advance."

**Line 461-466**

"Second, this study merely identifies the physical linkage between the interdecadal summer EAMBZ precipitation and the contemporaneous SST mode over the TIO basin from the tropical route. Nonetheless, the contemporaneous IOBM is not a predictor. According to many previous

studies (e.g., Wang et al., 2015; Li et al., 2023), the physical-based empirical model based on multiple predictors may better improve the forecast skill. Thus, it is urgent to find out more salient precursor signals of the lower boundary anomalies [e.g., sea ice (Han et al., 2021)] and figure out associated mechanisms for interdecadal EAMBZ precipitation changes to construct an effective prediction model."

**Reference:**

Wang, B., Xiang, B., Li, J., Webster, P.J., Rajeevan, M.N., Liu, J. and Ha, K.-J., 2015. Rethinking Indian monsoon rainfall prediction in the context of recent global warming. Nature Communications, 6(1): 7154.

Li, J., Zheng, C., Yang, Y., Lu, R. and Zhu, Z., 2023. Predictability of spatial distribution of pre-summer extreme precipitation days over southern China revealed by the physical-based empirical model. Climate Dynamics, 61(5): 2299-2316.

Han, T., Zhang, M., Zhu, J., Zhou, B. and Li, S., 2021. Impact of early spring sea ice in Barents Sea on midsummer rainfall distribution at Northeast China. Climate Dynamics, 57(3): 1023-1037.

And related to the above point, I think the discussion (Section 4) could be improved. At present, it summarises the results – which are they summarised again in the Conclusion – but gives little sense of how the results fit into existing knowledge. How does this work move the field forward? What questions arise from it?

**Reply:** Thanks for your constructive comments. We modified the Discussion Section in the raw manuscript into Section 3.4 "Results from CESM1 simulations", with the aim of confirming our proposed mechanisms based on the statistical results. Notably, we have improved the Section 3.4 in the revised manuscript, avoiding the repeated summary of the results. Please see **Line 376-389** in the revised version.

**Line 376-389**

"3.4 Results from CESM1 simulations

In this subsection, we use the pacemaker experimental data based on the ensemble mean of CESM1_IOPES and CESM1_LENS to validate our proposed mechanisms regarding the modulation of IOBM cooling on the interdecadal enhancement of summer EAMBZ precipitation. Considering the predominant role of southerly anomalies over the key monsoonal southerly domain, we therefore emphasize the low-level (850 hPa) atmospheric anomalies at interdecadal timescales tied to the IOBM-like SST cooling, as depicted in Fig. 10. We can observe a clearly anomalous cyclonic circulation around the northeast corner of TIO, accompanied by local positive precipitation anomalies and easterly anomalies that stretch from SWP to its northern flank, which are generally resembled those in the observation (Figs. 7b and 9b). In this circumstance, a similar "north-low–south-high" meridional seesaw pattern over the Northeast China–SWP sector can be formed to spark and sustain the enhanced EAMBZ precipitation in boreal summer (Fig. 10). In summary, by and large, the ensemble mean composite results can well reproduce the observed anomalous circulation and precipitation driven by IOBM-related SSTAs, confirming the crucial role of IOBM cooling in driving enhanced summer precipitation over EAMBZ at interdecadal timescales."

As for your concerned questions, please see **Line 452-459** in the revised version.

**Line 452-459**

"The following two points deserve further discussion. First, although results from CESM1_LENS and CESM1_IOPES can reasonably confirm our proposed physical pathway of how IOBM cooling exerts a distant modulation on the interdecadal enhancement of summer precipitation over EAMBZ, we can still notice the weakness of the model simulations. That is, positive precipitation anomalies around the northeast corner of TIO and the easterly anomalies exhibit weaker magnitudes compared to the observations (Fig. 10 vs. 7b and 9b). Besides, systematic biases exist regarding the simulated positions of the upper (lower) tropospheric divergence (convergence) and negative (positive) RWS anomalies (Fig. S6), manifesting themselves in the eastward displacement tendency in contrast to those around the northeast corner of the TIO (Fig. 8)."

Review of "Potential modulation of Indian Ocean basin mode on the interdecadal variations of summer precipitation over the East Asian monsoon boundary zone" by Jing Wang et al.

Many thanks for your constructive and valuable comments, which have greatly improved our manuscript.

This study looks into the link between the Indian Ocean Basin mode (IOBM) and the summer precipitation over the East Asian monsoon boundary zone (EAMBZ). The authors discusses, during cold phase of IOBM, the EAMBZ has enhanced precipitation. This is through the formation of an anomalous cycolonic circulation in the North-East corner of tropical Indian Ocean driving a North-low-South-high pattern taking place in the interdecadal timescales.

Overall, this study attempts to understand the features of precipitation over EAMBZ on interdecadal timescales. I would recommend this study for major revision and has to address below queries:

We have revised the manuscript based on your comments. The revisions are highlighted in red color in the revised manuscript. In the following, we summarize our point-by-point replies to your comments.

Major comments:

1. The title is not clear. It seems to not highlight that this study investigates how modulation of IOBM causes a a change in precipitation in the EAMBZ on interdecadal timescales. The title could be changed to "Modulation of Indian Ocean basin mode potentially drives the interdecadal variations of summer precipitation over the East Asian monsoon boundary zone". The authors ought to think on this and clarify.

**Reply:** Thanks for your insightful comments. The verb "modulate" seems to show a similar meaning of "drive". Considering your comments, we changed the title into "Role of Indian Ocean basin mode in driving the interdecadal variations of summer precipitation over the East Asian monsoon boundary zone", highlighting the driving role of IOBM. Please see **Line 1-3** in the revised version.

2. The study primarily provides mechanistic explanation on the link between IOBM and precipitation over EAMBZ during the cold phase only. The authors need to provide and discuss the processes during the IOBM warm phase and corresponding precipitation over EAMBZ, or is it that the authors discuss this somewhere and I have missed it.

**Reply:** Thanks for your valuable comments. We have discussed the processes during the IOBM warm phase and the corresponding precipitation over EAMBZ. Please see

**Line 372-374** in the revised version and **Figure S5** in the Supplementary File.

**Line 372-374**

"Notably, circulation and precipitation anomalies during the warm phase years of the IOBM (Fig. S5) highly mirror those tied to the IOBM cooling with opposite signs."

[Figure]

**Figure S5.** Composite anomalies of JJA-mean (a) Z400 (shading; m) and UV400 (vectors; m s$^{-1}$), (b) Z850 (shading; m) and UV850 (vectors; m s$^{-1}$), (c) <WVT> (vectors; kg m$^{-1}$ s$^{-1}$) and <WVT_div> (shading; $10^{-5}$ kg m$^{-2}$ s$^{-1}$), and (d) precipitation (mm month$^{-1}$) during the warm phase years of the IOBM. All variables are detrended and 11-year low-pass filtered. Areas with significant values of Z400, Z850, and <WVT_div> that exceed the 95% confidence level are stippled, respectively. Only vectors that are significant at the 95% confidence level are shown. The base period is 1901–2014. The warm phase years of the IOBM are selected based on the 0.5 standard deviations of the observed time-evolving SSTAs during the based period, as shown in Fig. 6b (blue line). The precipitation is derived from the CRU TS3.26 precipitation data; whilst other variables are from the 20CRv2c datasets.

3. I would urge the authors to make clear discussion on past studies (it is there already but somehow the present structure mingles them) which investigates the interannual variations of precipitation over EAMBZ. I recommend the literature review made in the "Introduction" to be more structured. Right now it reads unclear and less motivating to read through. Section 2 in the manuscript for instance, is very well structured and written. Another concern, far too many acronyms are used.

**Reply:** Thanks for your valuable comments. We have improved the "Introduction" section in the revised manuscript, avoiding the structure mingling issue. Specifically, in the third paragraph of the "Introduction", we highlighted the discussion on past studies, which investigates the interannual variations of precipitation over EAMBZ. Please see **Line 61-82** in the revised version.

Furthermore, we have removed the sparingly used acronyms and kept the frequently used acronyms in the revised manuscript. For easy reading and reviewing, we have included the "Glossary of acronyms" in the revised Supplementary File. Please see the Glossary of acronyms in the Supplement File.

4. IN addition, as the authors have discussed several processes that are interlinked and connects the IOBM and EAMBZ precipitation. It is necessary that they provide a schematic and/or flowchart showing the interlinkages and processes. This is would be very helpful for the readers.

**Reply:** Thanks for your valuable comments. We have included a schematic diagram showing the interlinkages and processes. Please see **Line 422-424** and **Figure 12** in the revised version.

**Line 422-424**

"As a summary of our major findings, Fig. 12 schematically synthesizes how IOBM-associated SST mode remotely drives the interdecadal precipitation fluctuations via a tropical route."

[Figure]

**Figure 12.** Schematic diagram showing how IOBM-related SST anomaly pattern drives the summer EAMBZ precipitation fluctuations at interdecadal timescales. Blue shading illustrates the IOBM cooling. Letter A (C) indicates the center of the anticyclonic (cyclonic) gyre anomaly.

Minor Comments:

Figure 1- Improve the resolution. It is a key figure.

**Reply:** Revised as suggested. We also uploaded a PDF format of Figure1, which shows the well resolution.

P2L82-83: I would urge the authors to be elaborate here.

**Reply:** Thanks for your helpful comments. Please see **Line 88-92** in the revised version.

**Line 88-92**

"For example, J. Wang et al. (2022) reported that the late spring (May) southeastern TP underwent wet conditions for 1928–1961 and 1989–2003, and experienced dry conditions preceding 1927, 1962–1988, and 2004 onwards. Si and Ding (2016) documented that East Asia experienced dry summers from the early 1920s to the 1940s, as well as wet summers from the late 1900s to the early 1920s, in the 1950s, and from the 1980s to the 1990s. Piao et al. (2021) found that the decadal-filtered summer precipitation over Northeast Asia underwent a sudden decease around the late 1990s."

P6L241: I also find dry summers around 1940. Authors mention about Si et al. 2021 here. Please write the context, since that paper looks into Northeast Asian summer monsoon and AMO.

**Reply:** Thanks for your helpful comments. Please see **Line 252-257** in the revised version for our modifications.

**Line 252-257**

"For example, EAMBZ experienced dry summers during the periods preceding 1927, 1939–1945, 1968–1982, and 1998–2010, but underwent wet summers during the periods of 1928–1938, 1946–1967, and 2011 onwards. Note that Si et al. (2021) explored the interdecadal variations of summer precipitation over northeast Asian, a domain that largely matches our focused EAMBZ domain. The observed major interdecadal fluctuation periods of EAMBZ precipitation are basically consistent with those suggested by Si et al. (2021), with dry summers around 1940."

P8L320: "localised atmospheric responses" Please explain.

**Reply:** Thanks for your comment. Please see **Line 333-337** in the revised version.

**Line 333-337**

"Moreover, there are striking suppressed precipitation around the northeast corner of the TIO domain (Fig. 7b), suggesting profoundly localized atmospheric responses (viz. the release of regional anomalous atmospheric cooling) to the warm TIO SSTAs. Note that corresponding to cold TIO SST years, there exist positive precipitation anomalies around the northeast corner of TIO,

suggesting the release of anomalous atmospheric heating (figure not shown)."

Moreover, your concerned "localised atmospheric responses" is tied to a low-level cyclonic in situ in terms of a typical Gill–Matsuno-type response. Please see **Line 361-366**.

**Line 361-366**

"One may ask how IOBM cooling induces the above-mentioned meridional seesaw pattern. Previously, we have revealed that negative SSTAs over TIO may exert remote interdecadal impacts through an atmospheric bridge, i.e., vigorous convective activities around the northeast corner of TIO (Figs. 7 and 8). In effect, there exists a low-level cyclonic anomaly in situ (Fig. 9b). Such cyclonic anomaly can be interpreted as a typical Gill–Matsuno-type response (Matsuno, 1966; Gill, 1980) to the regional anti-symmetric atmospheric heating caused by IOBM cooling with the coldest center located south of the equator, which is more clear within the lower levels (Fig. 9b)."

Equation5: The P-E model, please discuss the possible inabilities of this model. Also, is this model developed for this study (as mentioned in abstract) or does it follows from past studies such as Jeong et al. 2021. Please make this clear.

**Reply:** Thanks for your valuable comments. We have extended the discussion concerning the shortcomings of our proposed physical-based empirical model in Section 3.5. Please see **Line 411-416** in the revised version.

**Line 411-416**

"Although our proposed physical-based empirical model could confirm the concurrently intimately interdecadal relationship between IOBM and EAMBZ precipitation, we should acknowledge the shortcomings of the model. First, the amplitudes of the hindcast estimates are fairly lower, which cannot capture the extreme precipitation years (e.g., years around 1960; Fig. 11). Second, the simultaneous signal of IOBM cannot be served as a predictor for summertime EAMBZ precipitation variations. As such, this model inherently lacks the ability to predict the interdecadal EAMBZ precipitation anomalies in advance."

In addition, this model is developed for this study, and we only follows the method of Jeong et al. (2021). Please see **Line 405** in the revised version.

**Line 405**

"Following the method of Jeong et al. (2021),…".

---

## Author Response (AR2)

Dear Editor Dr Peer Nowack,

Many thanks for your hard editorial work. We also thank the two Reviewers for their constructive and valuable comments/suggestions. These comments/suggestions greatly improved our manuscript. According to the comments from the Editor decision, we have modified the manuscript based on Reviewer 1's comments/suggestions. In the online re-submission, we have included our point-by-point replies to these comments. Track changes are given in the revised manuscript. The following are our point-by-point replies to Reviewer 1's comments/suggestions.

We are looking forward to a further decision on the revised version.

Best regards,

Jing Wang, Yanju Liu*, Fei Cheng, Chengyu Song, Qiaoping Li, Yihui Ding, and Xiangde Xu

The revised manuscript has improved and the authors have addressed the comments. I thank the authors to include the schematic Figure 12 in the revised manuscript. However, yet at this revision as well I believe that the authors have not clearly stated what is new in this study and different from past studies. Ambiguity exists. I recommend for major revision, I have listed my comments below and request the authors to address them.

**Reply:** Many thanks for your constructive and valuable comments, which can greatly improve our manuscript. We have revised the manuscript based on your comments. The revisions are highlighted in track changes with red color in the second revised version. In the following, we summarize our point-by-point replies to your comments.

Major comments:

P3L100-103: The authors seek to address two questions 1) if there is interdecadal variations in the JJA EAMBZ precipitation? 2) if this is there then how IOBM and EAMBZ precipitation are interlinked? There are past studies such as Si et al. 2021(in their Figure 2) which showed the interdecadal fluctuations based on CRU-TS3.26 from 1900-2012 and also some station data from near Yangtze River which typically answers the first question this study is asking. This study uses the same CRU dataset but from 1901- 2014. Similarly, for the second question, past studies such as Zhang et al. 2018 did discuss and study the connection between IOBM and EAMBZ precipitation.

Although this study cite both Si et al. 2021 and Zhang et al. 2018, but the authors did not clarify what is new in this study compared to the past ones.

Zhang, Z., Sun, X. and Yang, X.-Q., 2018: Understanding the interdecadal variability of East Asian summer monsoon Understanding the Interdecadal Variability of East Asian Summer Monsoon Precipitation: Joint Influence of Three Oceanic Signals DOI: 10.1175/JCLI-D-17-0657.1

Si, D., Jiang, D., Hu, A. and Lang, X., 2021:Variations in northeast Asian summer precipitation driven by the Atlantic multidecadal oscillation DOI: 10.1002/joc.6912.

**Reply:** Thanks for your highly valuable comments. We have highlighted what is new in this study compared to the past ones.

For the publication Si et al. 2021 you emphasized (corresponds to the first question: 1) if there is interdecadal variations in the JJA EAMBZ precipitation?), please see **Line 50-55**, **Line 180-183,** and **Line 263-267** in the second revised version for the research innovation of this manuscript.

*Line 50-55*

*"It is essential to point out that although the EAMBZ domain largely overlaps the Northeast Asian area suggested by Si et al. (2021), the EAMBZ is defined from the perspective of the interaction between the mid-latitude westerly and the EASM [see Fig. 1 in Chen et al. (2021); also see the red box in Fig. 1 and associated description in Sect. 2.5.1], not from a geographical notion. Accordingly, the EAMBZ is a transitional climate zone between the EASM-controlled moist region and the westerly-dominated arid region over central Asia (Chen et al., 2010; Chen et al., 2018, 2021), stretching from the eastern flank of the TP to Mongolia and Northeast China."*

*Line 180-183*

*"Note that our focused EAMBZ domain differs from the Northeast Asian domain (29°–50°N, 108°–140°E) suggested by Si et al. (2021). Although they are extensively overlapped, the EAMBZ is located more westward and northward, and defined from the climatic system perspective, not from a pure geographical perspective."*

*Line 263-267*

*"Note that to some extent, the observed major interdecadal fluctuation periods of summertime EAMBZ precipitation are dissimilar from those tied to summertime Northeast Asian precipitation revealed by observations (1900–2012) from 11 local meteorological stations (Si et al., 2021), e.g. the above-normal precipitation over EAMBZ (Fig. 1e) vs. the below-normal precipitation over Northeast Asia around 1990 (Si et al., 2021; their Fig. 2a)."*

Furthermore, for the publication Zhang et al. 2018 you emphasized (corresponds to the second question: 2) if this is there then how IOBM and EAMBZ precipitation are interlinked?), please see **Line 317-324** in the second revised version for the research innovation of this manuscript, as you suggested again in the last comment of the Minor Comments.

*Line 317-324*

*"Many previous studies have substantiated that the IOBM can remotely modulate summer rainfall fluctuations over the mid-latitude Asia at interdecadal timescales (e.g., Zhang et al., 2018; S. Wang et al., 2022; Wu et al., 2022). Note that the existing studies primarily highlighted the impacts of IOBM on the summer rainfall variations over northwest portion of the mid-latitude Asia a (e.g., S. Wang et al., 2022; Wu et al., 2022). As for the work of Zhang et al. (2018), although this study focused the northeast portion of the mid-latitude Asia including the EAMBZ, it highlighted the combined roles of IOBM, AMO and PDO. In the present study, however, we identify that it is the IOBM that may exert profoundly simultaneous impacts on the interdecadal variations of the EAMBZ precipitation in boreal summer, which will be revealed subsequently."*

Abstract-L18-20: from observations, the interdecadal variations in the summer precipitation over EAMBZ has been shown by past studies and not a new finding. The authors could cite past studies or may include "as previously shown"…

**Reply:** Thanks for your helpful comments. Please see **Line 18-20** in the second revised version for our response.

*Line 18-20*

*"Observational evidence reveals that, similarly to previous studies, the EAMBZ precipitation featured prominent interdecadal fluctuations, e.g., with dry summers during the periods preceding 1927, 1939–1945, 1968–1982, and 1998–2010, and wet summers during the periods of 1928–1938, 1946–1967, and 2011 onwards."*

Abstract-L20-22: The IOBM connection to EAMBZ has been discussed by past studies such as Zhang et al. 2018, also cited by the authors. Here, it is important to clearly separate out the past findings and mention clearly the findings of this new study.

**Reply:** Thanks for your helpful comments. Please see **Line 20-22** in the second revised version for our response, highlighting the independent modulation role of the IOBM on the summertime EAMBZ precipitation at interdecadal timescales.

*Line 20-22*

*"Further analyses identify that the Indian Ocean basin mode (IOBM) is a significant oceanic forcing responsible for the interdecadal variations of the EAMBZ precipitation, playing an independent and critical modulation role."*

Abstract-L26-28: The physical-empirical model has its own limitations, so again clarifying that as included by the authors in this revised manuscript.

**Reply:** Thanks for your insightful comments. Please see **Line 27-29** in the second revised version for our response.

*Line 27-29*

*"For this reason, a physical-empirical model for the EAMBZ precipitation is developed in terms of the IOBM cooling. Despite the fact that the extreme summer EAMBZ precipitation cannot be captured by this model, it can still well capture its interdecadal fluctuations and reflect their steady relationship."*

Minor Comments:

P2L61-182: Is this discussion on the interannual variability in the summer EAMBZ precipitation really needed? The authors may want remove this paragraph.

**Reply:** Thanks for your question. However, we suggest that a review of the scientific advances regarding the physical mechanisms responsible for the interannual variability of summer EAMBZ precipitation is imperative. Such review is an organic part in this study. On one hand, such discussion you mentioned could tell the readers current knowledge gap about the summertime rainfall variability over the EAMBZ. That is, most existing pertinent studies focused on the interannual variability, paying

less attention to the interdecadal variability. On the other hand, the methodologies within the physical mechanisms responsible for the interannual variability of summer EAMBZ precipitation are helpful to unravelling the mechanisms tied to the interdecadal variability of summer EAMBZ precipitation in this manuscript. Therefore, it is reasonable for us to keep this paragraph.

P3L90: Please replace "as well as" to "while"

**Reply:** Revised as suggested. Thank you. Please see **Line 93** in the second revised version.

*Line 93*

*"..., while wet…"*

The authors write "oceanic interdecadal signals" in P3L92-93 and in the next sentence P3L95, they say referring to same as "interdecadal oceanic forcing". Generally, "signal" refers to response to some forcing. Please clarify.

**Reply:** Thanks for your insightful comments. We totally agree that "signal" refers to response to some forcing. To clarify this and highlight the forcing role of SST anomalies, the "oceanic interdecadal signals" has been changed into "interdecadal oceanic forcings" for consistency. Please see **Line 96** in the second revised version.

*Line 96*

*"The interdecadal oceanic forcings…"*

Similarly, in P6L214: Please rephrase "external radiative forcing signals". I believe the authors mean "the radiative forcing due to external perturbations such as GHGs"

**Reply:** Thanks for your constructive comments. Rephrased as suggested. Please see Line 219-220 in the second revised version.

*Line 219-220*

*"radiative forcing due to external perturbations such as greenhouse gases"*

P6L215-219: This is not clear. Taking an ensemble mean removes the internal variability and gives an estimate of forced response. So taking the ensemble mean from both CESM1_LENS and CESM1_IOPES would remove the variability. Please clarify.

**Reply:** Thanks for your constructive comments. We have clarified your concerns. Please see **Line 220-229** in the second revised version.

*Line 220-229*

*"Furthermore, the 10-member ensemble mean results in CESM1_IOPES contain the responses to both the time-evolving radiative forcing due to external perturbations and the restored observed time-varying SSTAs over the above broader TIO domain (Yang et al., 2020). Note that though the ozone forcing data used in CESM1_IOPES differ from those in CESM1_LENS, the differences in the corresponding simulated tropical and extratropical climates were indistinguishable (e.g., Schneider et al., 2015; Schneider and Deser, 2018; Zhang et al., 2019; Yang et al., 2020). Therefore, by subtracting the CESM1_LENS ensemble mean from the CESM1_IOPES ensemble mean (i.e., removing the shared radiative forcing described above), we can obtain the response of the climate system to the internal variability stemming from the time-varying SSTAs over the specific TIO, isolating the intrinsic climate variability driven by TIO SSTAs through excluding the impacts of the time-evolving external radiative forcing. More details about CESM1_LENS and CESM1_IOPES can be found in Kay et al. (2015) and Yang et al. (2020), respectively."*

*Reference:*

*Kay, J.E., Deser, C., Phillips, A., Mai, A., Hannay, C., Strand, G., Arblaster, J.M., Bates, S.C., Danabasoglu, G., Edwards, J., Holland, M., Kushner, P., Lamarque, J.F., Lawrence, D., Lindsay, K., Middleton, A., Munoz, E., Neale, R., Oleson, K., Polvani, L. and Vertenstein, M., 2015. The community earth system model (CESM) large ensemble project: a community resource for studying climate change in the presence of internal climate variability. Bulletin of the American Meteorological Society, 96(8): 1333-1349.*

*Schneider, D.P. and Deser, C., 2018. Tropically driven and externally forced patterns of Antarctic sea ice change: reconciling observed and modeled trends. Climate Dynamics, 50(11): 4599-4618.*

*Schneider, D.P., Deser, C. and Fan, T., 2015. Comparing the impacts of tropical SST variability and polar stratospheric ozone loss on the southern ocean westerly winds. Journal of Climate, 28(23): 9350-9372.*

*Yang, D., Arblaster, J.M., Meehl, G.A., England, M.H., Lim, E.-P., Bates, S. and Rosenbloom, N., 2020. Role of tropical variability in driving decadal shifts in the Southern Hemisphere summertime eddy-driven jet. Journal of Climate, 33(13): 5445-5463.*

*Zhang, L., Han, W., Karnauskas, K.B., Meehl, G.A., Hu, A., Rosenbloom, N. and Shinoda, T., 2019. Indian Ocean warming trend reduces Pacific warming response to anthropogenic greenhouse gases: An interbasin thermostat mechanism. Geophysical Research Letters, 46(19): 10882-10890.*

P8L307-308: At this point please explain here how this study is different from the past studies looking at the connection between IOBM and EAMBZ precipitation.

**Reply:** Thanks for your helpful comments. Explained as suggested. Please see **Line 317-324** in the second revised version.

*Line 317-324*

*"Many previous studies have substantiated that the IOBM can remotely modulate summer rainfall fluctuations over the mid-latitude Asia at interdecadal timescales (e.g., Zhang et al., 2018; S. Wang et al., 2022; Wu et al., 2022). Note that the existing studies primarily highlighted the impacts of IOBM on the summer rainfall variations over northwest portion of the mid-latitude Asia a (e.g., S. Wang et al., 2022; Wu et al., 2022). As for the work of Zhang et al. (2018), although this study focused the northeast portion of the mid-latitude Asia including the EAMBZ, it highlighted the combined roles of IOBM, AMO and PDO. In the present study, however, we*

*identify that it is the IOBM that may exert profoundly simultaneous impacts on the interdecadal variations of the EAMBZ precipitation in boreal summer, which will be revealed subsequently."*

---

## Author Response (AR3)

Dear Editor Dr Peer Nowack,

Many thanks for your hard editorial work again. We also thank Reviewer 2 for the constructive comments regarding the connection to aspects of previous work. According to the comments from the Editor decision, we have thoroughly addressed Reviewer 2's concerns. In the online re-submission, we have included our point-by-point replies to Reviewer 2's concerns. Please see track changes in the third revised manuscript for our responses.

We are looking forward to a further decision on the revised version.

Best regards,

Jing Wang, Yanju Liu*, Fei Cheng, Chengyu Song, Qiaoping Li, Yihui Ding, and Xiangde Xu

**The following are our point-by-point replies to Reviewer 1's concerns.**

The authors have positively addressed the comments. I thank the authors to clarify that their EAMBZ domain is different from that of Si et al. 2021. However, I would say that one can still infer the same conclusion about the interdecadal variations in the JJA EAMBZ precipitation from Si et al 2021.

**Reply:** Many thanks for your constructive comments. We have clearly stated that the EAMBZ domain in the current study is different from that of Si et al. 2021. Therefore, we saw distinctive conclusions about the interdecadal variations in the JJA EAMBZ precipitation, which are different from the work of Si et al 2021.

Please see **Line 267-271** in the third revised version for the related contrastive analysis.

**Line 267-271**
"Note that to some extent, the observed major interdecadal fluctuation periods of summertime EAMBZ precipitation are dissimilar from those tied to summertime Northeast Asian precipitation revealed by observations (1900–2012) from 11 local meteorological stations (Si et al., 2021), e.g. the above-normal precipitation over EAMBZ (Fig. 1e) vs. the below-normal precipitation over Northeast Asia around 1990 (Si et al., 2021; their Fig. 2a)."

[Figure]

Fig. 1e Normalized time series of the JJA-mean EAMBZ precipitation index (IEAMBZP) (red line) and associated first principal component (PC1) (blue line), with the number denoting the temporal correlation coefficient (TCC) between the corresponding time series. Precipitation data are detrended and 11-year low-pass filtered.

[Figure]

FIGURE 1 Distribution of the 11 observationstations(black dots) over Northeast Asia

Fig. 2a Time series of summer precipitation (unit: mm) over (a) Northeast Asia from 1900 to 2012, averaged for the 11 stations in Northeast Asia (see the figure to the right). The filling curve is the decadal-filtered values.

(Fig. 2a was copied from **Si et al., 2021**)

Also, to my next question about Zhang et al 2018 which investigated the connection between IOBM and EAMBZ precipitation, the authors responded that Zhang et al 2018 studied combined influence of IOBM, AMO as well as PDO on EAMBZ precipitation while the current study highlights the primary role of IOBM. However, except in L320-324 in this version of the manuscript, I do not find this message in the manuscript. Reading the abstract as well, one will not find this information.

Hence, I would request the authors to rewrite/restructure to highlight these two major points both in the abstract and the manuscript I) a different domain compared to Si et al 2021 II) relative to past studies such as Zhang et al. 2018, this study shows the primary role of IOBM. At this stage I would recommend for a major revision.

**Reply:** Many thanks for your helpful comments. We have restructured the Abstract and the final Section "Conclusions and discussion" to highlight the two issues of the different domain compared to Si et al 2021, as well as the role of IOBM.

Please see **Line 18-19** and **Line 22-23** in the **Abstract** for the third revised version.

*Revised Abstract*

"Based on long-term observational and reanalysis datasets from 1901 through 2014, this study investigates the characteristics and physical causes of the interdecadal variations in the summer precipitation over the East Asian monsoon boundary zone (EAMBZ), which is a peculiar domain defined from the perspective of the interplay between climatic systems (i.e., mid-latitude westerly and East Asian summer monsoon). Observational evidence reveals that, similarly to previous studies, the EAMBZ precipitation featured prominent interdecadal fluctuations, e.g., with dry summers during the periods preceding 1927, 1939–1945, 1968–1982, and 1998–2010, and wet summers during the periods of 1928–1938, 1946–1967, and 2011 onwards. Further analyses identify that, amongst the major interdecadal oceanic forcings (e.g., Atlantic multidecadal oscillation and Pacific decadal oscillation), the Indian Ocean basin mode (IOBM) is a significant oceanic forcing responsible for the interdecadal variations of the EAMBZ precipitation, playing an independent and critical modulation role. When the cold phase of the IOBM occurs, an anomalous cyclonic circulation is excited around the northeast corner of the tropical Indian Ocean, which further induces a "north-low–south-high" meridional seesaw pattern over the Northeast China–subtropical western Pacific (SWP) sector. Such seesaw pattern is conducive to the enhanced EAMBZ precipitation through linking favorable environments for the transportation of water vapor from the SWP and the convergence over EAMBZ at interdecadal timescales. For this reason, a physical-empirical model for the EAMBZ precipitation is developed in terms of the IOBM cooling. Despite the fact that the extreme summer EAMBZ precipitation cannot be captured by this model, it can still well capture its interdecadal fluctuations and reflect their steady relationship. The key physical pathway connecting the IOBM cooling with the interdecadal variations of the summer EAMBZ precipitation is supported by the numerical results based on the large ensemble experiment and the Indian Ocean pacemaker experiment. Our findings may provide new insights into the understanding of the causes of the interdecadal variations in the summer EAMBZ precipitation, which may favor the long-term policy decision making for the local hydrometeorological planning."

Please also see **Line 439-440** and **Line 457-458** in the **Conclusions and discussion** section in the third revised version.

**Line 439-440**

"EAMBZ is a peculiar domain defined from the perspective of the interplay between climatic systems (i.e., mid-latitude westerly and EASM). …"

**Line 457-458**

"We further identify that, amongst the major interdecadal oceanic forcings (e.g., Atlantic multidecadal oscillation and Pacific decadal oscillation), …"

Reference:

Si, D., Jiang, D., Hu, A. and Lang, X., 2021: Variations in northeast Asian summer precipitation driven by the Atlantic multidecadal oscillation DOI: 10.1002/joc.6912.

Zhang, Z., Sun, X. and Yang, X.-Q., 2018: Understanding the interdecadal variability of East Asian summer monsoon Understanding the Interdecadal Variability of East Asian Summer Monsoon Precipitation: Joint Influence of Three Oceanic Signals DOI: 10.1175/JCLI-D-17-0657.1.